

# Quantification and mitigation of bottom trawling impacts on sedimentary organic carbon stocks in the North Sea

Lucas Porz[1], Wenyan Zhang[1], Nils Christiansen[1], Jan Kossack[1], Ute Daewel[1], Corinna Schrum[1,2]

[1]Institute of Coastal Systems, Helmholtz-Zentrum Hereon, Max-Planck-Strasse 1, 21502 Geesthacht, Germany
[2]Institute of Oceanography, Center for Earth System Research and Sustainability, Universität Hamburg, Bundesstrasse 53, 20146 Hamburg, Germany

Correspondence to: Lucas Porz (lucas.porz@hereon.de)

**Abstract.** The depletion of sedimentary organic carbon stocks by use of bottom-contacting fishing gear and the potential climate impacts resulting from remineralization of the organic carbon to $CO_2$ have recently been heavily debated. In this study, a 3D coupled numerical ocean-sediment-macrobenthos model is used to quantify the impacts of bottom trawling on organic carbon and macrobenthos stocks in North Sea sediments. Using available information on vessel activity, gear components and sediment type, we generate daily time series of trawling impacts and simulate six years of trawling activity in the model, as well as four management scenarios in which trawling effort is redistributed from areas inside to areas outside of trawling closure zones. North Sea sediments contained 552.2±192.4 kt less organic carbon and 13.6±2.6% less macrobenthos biomass in the trawled simulations than in the untrawled simulations by the end of each year. The organic carbon loss is equivalent to aqueous emission of 2.0±0.7 Mt $CO_2$ each year, half of which is likely to accumulate in the atmosphere on multi-decadal timescales. The impacts were elevated in years with higher levels of trawling pressure and vice versa. Results showed high spatial variability, with a high loss of organic carbon due to trawling in some areas, while organic carbon content increased in nearby untrawled areas following transport and redeposition. The area most strongly impacted was the heavily trawled and carbon-rich Skagerrak. Simulated trawling closures in planned Offshore Wind Farms and Core Fishing Grounds had negligible effects on net sedimentary organic carbon, while closures in Marine Protected Areas had a moderate positive impact. The largest positive impact arose for trawling closures in Carbon Protection Zones, which were defined as areas where organic carbon is most vulnerable to disturbance. In that scenario, the net impacts of trawling on organic carbon



and macrobenthos biomass were reduced by 29% and 54%, respectively. These results demonstrate that carbon protection and habitat protection can be combined without requiring a reduction in net fishing
effort.

## 1.  Introduction

Bottom trawling, a fishing method whereby vessels drag weighted nets along the seabed to catch bottom-dwelling animals, is a major human disruption of the seabed. Chronic bottom trawling has been shown to depreciate ecological seabed habitats (Hiddink et al., 2017; Eigaard et al., 2017; Sciberras et al., 2018),
alter biogeochemical fluxes (van de Velde et al., 2018; Tiano et al., 2019; Bradshaw et al., 2021; Morys et al., 2021), and influence seabed morphology (Palanques et al., 2014; Puig et al., 2015; Oberle et al., 2016a; Amoroso et al., 2018; Porz et al., 2023). Recent geospatial modeling studies have estimated regional and global aqueous $CO_2$ emissions resulting from bottom trawling-induced remineralization of sedimentary organic carbon (OC), with some authors proposing seabed protection as an effective climate
protection measure (Luisetti et al., 2019; Sala et al., 2021; Black et al., 2022; Epstein and Roberts, 2022; Jankowska et al., 2022; Muñoz et al., 2023). Whereas such approaches consider the excess resuspension of the top sediment layer by bottom trawls, little is known about the transport and fate of the resuspended material, nor about the large-scale impacts of other physical and biogeochemical interactions between bottom trawls and OC (Table 1). As a result, the overall magnitude of bottom trawling impacts on carbon
budgets is still debated (Luisetti et al., 2020; Hilborn and Kaiser, 2022; Epstein et al., 2022; Hiddink et al., 2023).

Despite their recognized detrimental effects on seabed integrity, bottom trawling has not been considered in the designs of many marine management strategies such as the European Water Framework Directive (McLaverty et al., 2023). Efforts to mitigate seabed destruction typically call for an exclusion of trawling
in protected areas and are aimed at seafloor habitat conservation, often favoring sandy or hard bottoms such as reefs, and seldom consider carbon impacts. Similarly, other marine spatial management strategies such as offshore wind farm development have primarily targeted shallower, sandy bottoms. An exclusion of trawling in those areas may lead to an increased impact on sedimentary carbon if trawling effort is forced to relocate to muddier areas, which typically contain more OC (Smeaton and Austin, 2022). Such



relocation effects need to be resolved if marine spatial plans considering carbon protection are to have a sound scientific basis.

The North Sea, a shallow epicontinental shelf sea in the Northeast Atlantic, has been subject to chronic bottom trawling ("trawling" in the following) for more than a century (Thurstan et al., 2010). Though trawling effort in the North Sea has decreased during the past 20 years (ICES, 2017), it remains among
the most intensely trawled areas globally (Amoroso et al., 2018). While most of the North Sea's seafloor is covered by relict sands which do not accumulate sediment or OC at significant rates, muddy hotspots of deposition do exist and are located primarily in topographic depressions or areas otherwise shielded from waves and erosional currents: The Norwegian Trench and Skagerrak, Fladen Ground, Oyster Ground, as well as smaller patches in the German Bight and at the UK's coasts (Fig. 2b). Trawling effort
in the North Sea is spatially heterogeneous, with some areas of the seafloor contacted more than ten times each year on average, while other areas are completely untrawled (Fig. 3). Trawling effort is elevated at some depositional areas, most notably in parts of the Skagerrak, the edge of the Norwegian Trench and Fladen Ground. While several studies have addressed short-term, local responses of various North Sea sediments and benthos to trawling-induced disturbances (e.g. de Groot, 1984; Rijnsdorp et al., 2020;
Bruns et al., 2023), the overall impacts of trawling on the sedimentary OC budget of the North Sea remains unknown.

In this study, we use a 3D numerical model of the North Sea to simulate hydrodynamics, sediment dynamics, macrobenthos functions and trawling impacts (Fig. 1). We account for four major trawling impacts: 1. Resuspension and associated remineralization, 2. Redistribution by transport and redeposition,
3. Macrobenthos depletion and associated changes of bioturbation and respiration, and 4. Mechanical vertical sediment mixing due to penetrating gear components (anthroturbation). This allows an estimate of the large-scale impacts of trawling on the North Sea's macrobenthos biomass and sedimentary OC sequestration capacity, i.e. the amount of OC removed from the carbon cycle due to sedimentation over time. In addition, we simulate the potential impacts of four different marine management scenarios by
redistributing the trawling effort within areas where trawling would be prohibited in those scenarios.



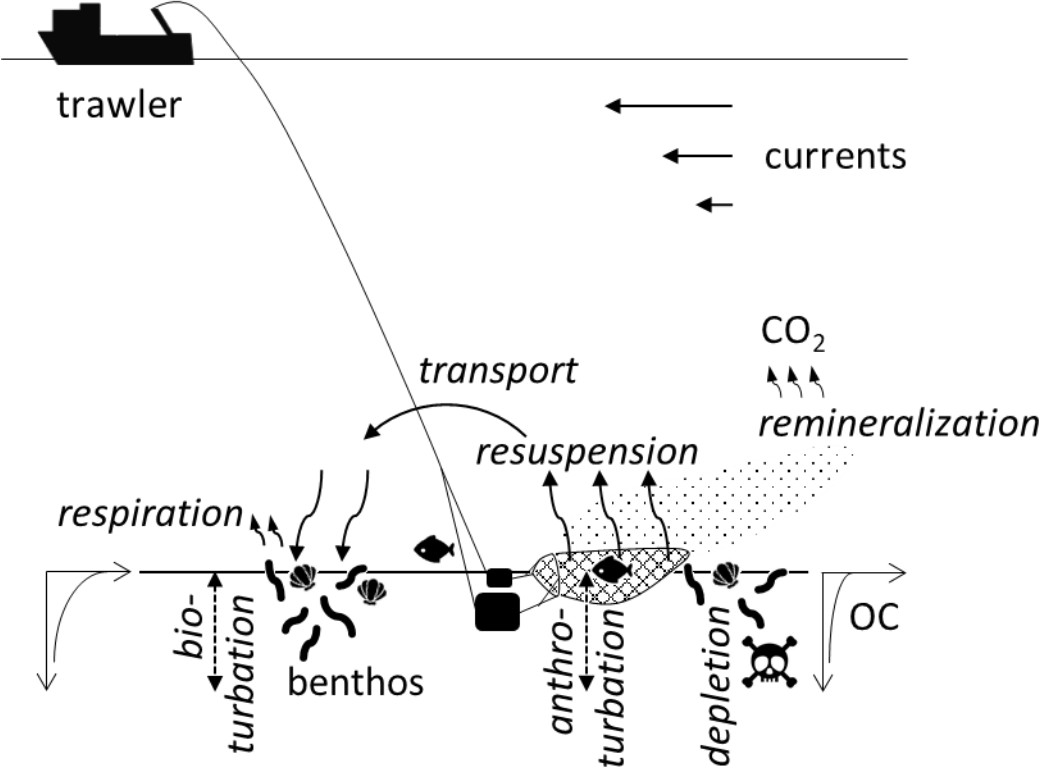

Figure 1. Interaction of trawlers with sedimentary organic carbon. Processes considered in this modeling study are in italics.




Table 1. Possible bottom trawling impacts on sedimentary carbon. The order indicates the immediacy with which the effects manifest in the sedimentary OC budget. Higher-order effects are not necessarily smaller in magnitude, but rather greater in complexity and therefore more difficult to quantify. Mechanisms and their possible impacts indicate whether they are likely to increase (+) or decrease (−) OC sequestration. Bold font indicates impacts considered in this study. References include studies that have addressed the respective impacts.

| Order | Effect/mechanism | Possible impacts on OC sequestration | References |
|---|---|---|---|
| 1st | Sediment erosion/resuspension | **Increased remineralization (−)** | de Borger et al. (2021b) Morys et al. (2021) Sala et al. (2021) van de Velde et al. (2018) Paradis et al. (2021) |
| | Physical churning by penetrating gear components (Anthroturbation) | **Increased sediment mixing (+)** | de Borger et al. (2021b); Bunke et al. (2019) van de Velde et al. (2018) Paradis et al. (2021) Duplisea et al. (2001) Oberle et al. (2016b) |
| | | Increased porewater fluxes (−) | |
| 2nd | Depletion of benthos | **Reduced benthic respiration (+)** | Tiano et al. (2019) |
| | | **Reduced bioturbation (−)** | |
| | Lateral transport of resuspended material | **Increased deposition in deeper areas (+)** | Paradis et al. (2019) Paradis et al. (2022) |
| 3rd | Nutrient resuspension | Increased production in shallow areas (+) | Dounas et al. (2007) |
| | Increased turbidity | Reduced production in shallow areas (−) | - |
| 4th and higher | Bottom oxygen depletion due to remineralization | Reduced bioturbation (−) | - |
| | | Reduced remineralization (+) | |
| | Removal of macrobenthos | top-down food web feedbacks (−)/(+) | - |



## 2. Materials and methods

### 2.1. Numerical model

The coupled numerical modeling system used in this study comprises three parts: The Semi-implicit Cross-scale Hydroscience Integrated System Model (SCHISM; Zhang et al., 2016) for hydrodynamics, a sediment transport and morphodynamics model (MORSELFE, Pinto et al., 2012) based on the Community Sediment Transport Model (CSTM; Warner et al., 2008) for sediment dynamics, and the Total Organic Carbon-Macrobenthos Interaction Model (TOCMAIM; Zhang and Wirtz, 2017) for 100 interactions of OC and macrobenthos in the sediment.

The hydrodynamic setup of SCHISM is based on that of Kossack et al. (2023), which has been validated for the North Sea. The hydrodynamic model domain encompasses the entire Northwestern European Shelf, including the Baltic Sea and extending past the shelf break into the North Atlantic (Fig. **2**a), allowing internal circulation patterns to emerge within the North Sea that are not imposed by the boundary 105 conditions alone. The horizontal resolution of the unstructured computational grid increases from 15−20 km in the North Atlantic to ~10 km in the North Sea and to a few kilometers near the coast. The atmospheric hindcast simulation coastDat-3 (Geyer, 2017) is used for atmospheric forcing.

The initialization of seabed sediment in the model is restricted to the study area in the North Sea (see Fig. 2a). Six sediment classes are defined, three of which represent inorganic particles (sand, silt, and clay), 110 and three of which represent OC pools of different bioavailability and degradation rates (fresh, semi-labile and refractory). The inorganic sediment fractions are initialized according to sediment maps of Bockelmann et al. (2018). Organic sediment and macrobenthos biomass are initialized from multi-decadal (1950−2000) TOCMAIM simulations (Zhang et al., 2021), where the model domain has been extended from the Southern North Sea to the entire North Sea for this study. The model TOCMAIM has 115 been extensively validated against measurements of OC and macrobenthos in the North Sea (Zhang and Wirtz, 2017; Zhang et al., 2019; Zhang et al., 2021). Macrobenthos grows and declines according to OC availability and temperature, as given by an ecosystem model (Daewel and Schrum, 2013), and diffusively mixes the sediment fractions in the sediment bed vertically through bioturbation. Bioturbation diffusion coefficients are scaled according to biomass.



The seabed is discretized into 30 vertical layers with an initial thickness of 1 cm per layer. During simulation, sediment layer thicknesses and fractions are adjusted dynamically based on erosion, deposition and mixing. Erosion occurs when the bottom shear stress calculated in the hydrodynamic model exceeds a critical value. Eroded sediment is treated as a sinking tracer in the hydrodynamic model that can be mixed, circulated and redeposited. Detailed parameter settings for the sediment model are

listed in **Table A1**. In the analysis of the model results, the entire model domain is considered when budgeting the total OC mass. In this way, OC that has been transported and deposited outside of the study area by currents is accounted for.

In order to gauge the inter-annual variability due to hydrodynamics and level of trawling pressure, we simulate six consecutive years starting from 2000. That time period saw a moderate increase in demersal

fish landings during the first years, followed by a sharp decrease in the later half, after which the levels have remained similar until recent years (ICES, 2023b). The model is re-initialized at the end of each year, such that the results of each year remain comparable among each other and their differences can be attributed solely to inter-annual differences in external forcing, i.e., atmospheric conditions, river loads, oceanic boundaries and  trawling pressure.



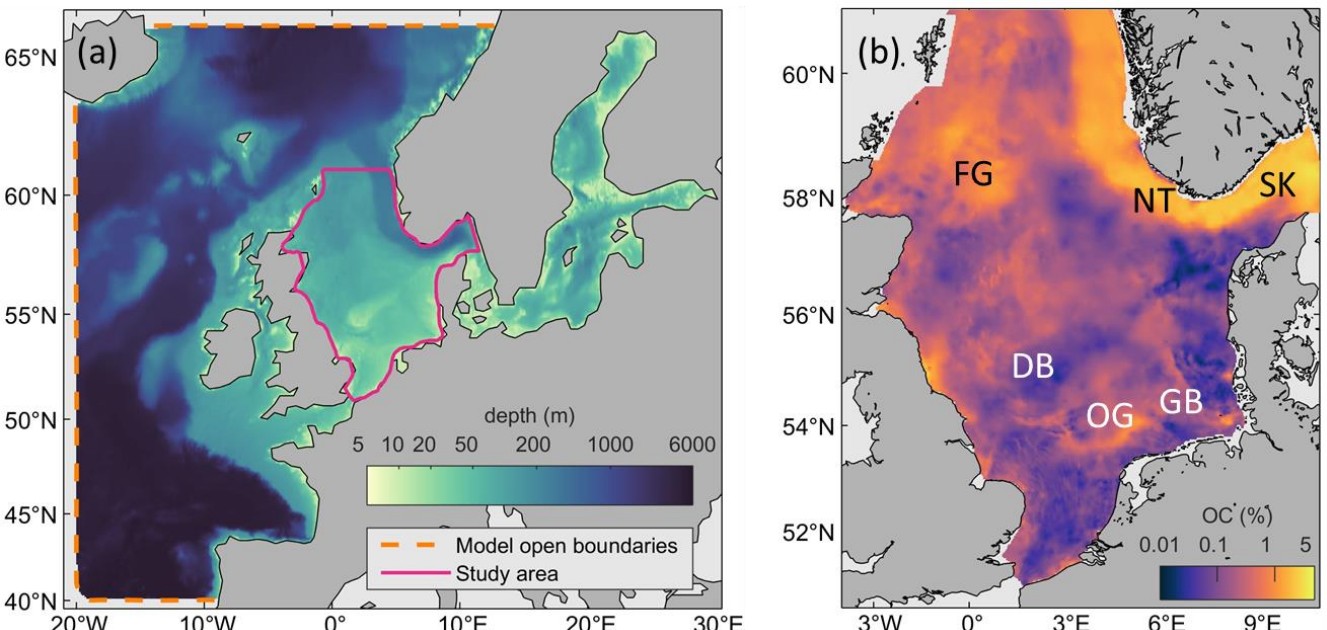

Figure 2. Study area. (a) Model domain and bathymetry with open boundaries and study area. (b) Sediment organic carbon in the North Sea according to Bockelmann et al. (2018). FG=Fladen Ground, NT=Norwegian Trench, SK=Skagerrak, DB= Dogger Bank, OG=Oyster Ground, GB=German Bight.

### 2.2. Synthesis of bottom trawling activity and impacts

We generate daily time series of trawling effort from the GlobalFishingWatch dataset at 0.1°×0.1° resolution (GFW, Kroodsma et al., 2018) for the North Sea within limits of −4°E to 12°E and 50° to 62°N. The GWF data contains daily vessel locations and fishing effort, as well as estimates of vessel power, length and class. A comparison of the fishing effort of vessels classified as "trawlers" by GFW to bottom trawling effort according to annual governmental surveys (ICES, 2019) for the year 2017 shows overall agreement both in terms of total trawling hours and spatial distribution (Fig. 3). All trawlers in the GFW data are considered bottom trawlers for the purpose of this study. Trawled hours in the GFW data within the study area for the years 2012 to 2017 are lower than bottom trawled hours in ICES (2019) by 81, 33, 22, 17, 16, and 10%, respectively. This indicates that both datasets are converging as more vessels are included in the GFW data, and we consider the GFW data before 2015 unreliable for our purpose. For the simulation period of 2000–2005, the daily fields of 2015–2020 are averaged and scaled according to annual historical landings of demersal fish reported in ICES (2017) for the years 1950 to 2015, using the year 2017 as a baseline. According to ICES (2017), landings of demersal fish rose by a factor of three in





1970–2000, after which the landings decreased to nearly the pre-1970s level. In total, the GFW daily fished hours averaged over 2015–2020 are about 10% higher than those of 2017, somewhat mitigating
the 10% lower effort in the GFW data. This approach assumes that the spatial and seasonal patterns of trawling effort have not changed significantly over time, which is supported by historical data (Couce et al., 2020).

The GFW data does not distinguish between specific trawled gear types. As vessels in the GFW data typically operate within a 1°-radius, a gear type is assigned to each vessel at the vessel's average position
according the dominant métier defined by Eigaard et al. (2016; data in ICES, 2019). A métier groups fishing trips by gear type and target species, and vessels operating in the same métier are expected to have similar impacts on the seabed per unit area contacted. Fourteen métiers have been defined, of which eight are otter trawlers, three are beam trawlers, two are demersal seines and one is dredges. In this study, only the trawler groups are considered, because reliable estimates for the impacts of the other bottom-
contacting gear types (demersal seines and dredges) are not available. Overall, demersal seines and dredges together have recently made up less than 10% of fishing hours in the North Sea (ICES, 2019).

The empirical expressions in Eigaard et al. (2016) are applied to estimate gear widths from vessel length or vessel power, as well as average towing speeds and the length proportions of gear components for each métier. To avoid extrapolating outside of the data gathered by Eigaard et al. (2016), vessel size and engine
power are limited to the maximum values of their data points for each métier, thus preventing excessively large gear widths which would otherwise occur in less than 5% of all vessels in the dataset.



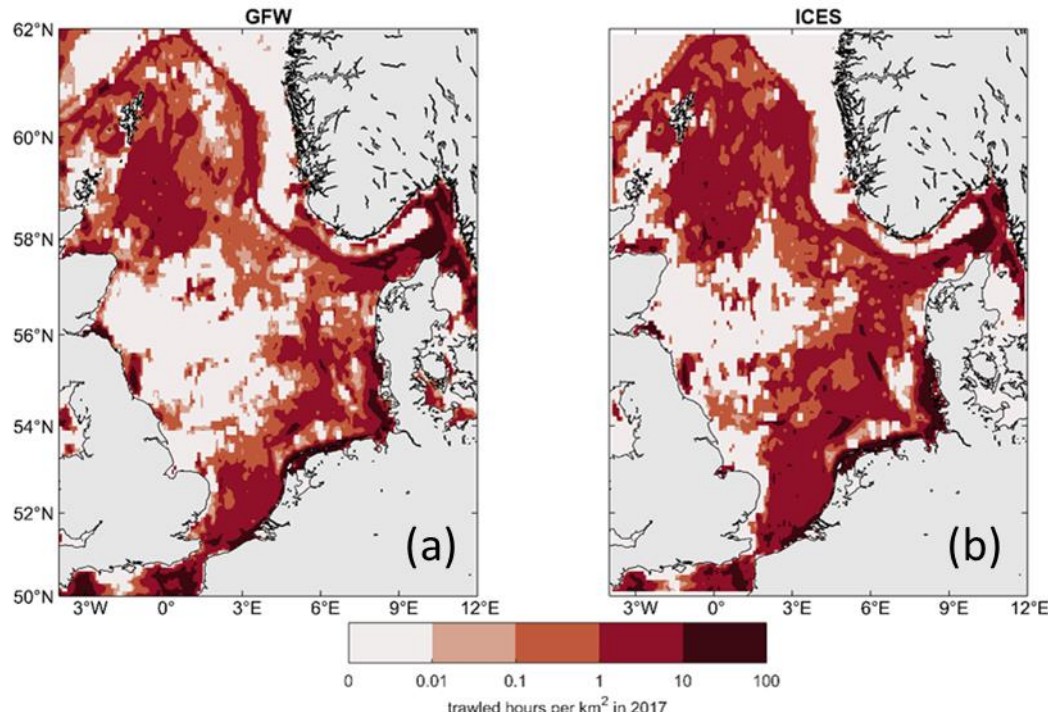

Figure 3. Comparison of trawling effort data. (a) Trawled hours according to GFW (Kroodsma et al., 2018) and (b) bottom trawled hours according to ICES (2019) for the year 2017, respectively. Effort is expressed as trawled hours per area. The data in ICES (2019) was resampled to a 0.1°×0.1° grid for better visual comparability. Total trawled hours in (a) are 10% lower than in (b).

### 2.2.1. Resuspension rates

In order to estimate the resuspension rate of each trawler, we follow O'Neill and Ivanović (2016), who demonstrated that sediment entrained behind towed bottom-contacting fishing gear is related to the hydrodynamic drag of the gear components and to the seafloor sediment type. Their empirical formula for sediment mobilized per contacted area (in kg m$^{-2}$) is:

$$m = 2.602 \cdot s_\mathrm{f} + 1.206 \cdot 10^{-3} \cdot H_\mathrm{d} + 1.231 \cdot 10^{-2} \cdot s_\mathrm{f} \cdot H_\mathrm{d}, \qquad (1)$$

where $s_\mathrm{f}$ is the silt content of the seabed and $H_\mathrm{d}$ (in N m$^{-1}$) is the hydrodynamic drag per meter width of the gear component. Estimates for $H_\mathrm{d}$ are taken from literature (see Appendix B for details) and the silt fraction of the sediment is assigned according to the surface sediment map of Bockelmann et al. (2018). The resuspension rate (in kg s$^{-1}$) of a vessel's total gear can be determined as



$$E_{\text{trawl}} = v \cdot \sum_j m_j \cdot W_j, \tag{2}$$

where $W$ (in m) is the width of gear component $j$ and $v$ (in m s$^{-1}$) is the vessel speed. The width of nets is assumed to equal the width of the ground gear for all gear types. Resuspension per area contacted is limited to 6 kg m$^{-2}$ in order to avoid excessive resuspension outside of reported values (Oberle et al., 2016a). In 2017, the modeled resuspension per area contacted calculated in this way is 2.1±1.6 kg m$^{-2}$, which is well within the range of reported values. The associated average resuspension rate per vessel is 213 kg s$^{-1}$.

Daily resuspension rates per area (in kg m$^{-2}$ s$^{-1}$) are calculated for each model grid cell as the sum of impacts of all vessels:

$$e_{\text{daily}}^k = \sum_i E_{\text{trawl}}^{i,k} \cdot \frac{T_{\text{trawl}}^{i,k}}{1 \text{ day} \cdot A_k}, \tag{3}$$

where $T_{\text{trawl}}$ is the duration of time during which vessel $i$ was trawling in grid cell $k$, and $A$ is the grid cell area.

### 2.2.2. Swept area ratios

We estimate the daily swept area ratio (SAR), which represents the portion of an area of seabed contacted by trawling gear, in three depth intervals within the seabed: 0–2 cm, 2–5 cm, and 5–10 cm. The depths to which the individual gear components penetrate in sandy and muddy sediment are listed in Table 2 according to data in Eigaard et al. (2016).





Table 2. Penetration depths of different gear components used in the model. Depth levels 1, 2 and 3 correspond to penetration depths of 2 cm, 5 cm, and 10 cm, respectively. Ground gear is separated into a surface and a subsurface component based on the métier using the ratios in Eigaard et al. (2016). Areas with a silt fraction ≥10% are designated as mud, and the rest is designated as sand.

| Gear component | Penetration depth level | |
|---|---|---|
| | Sand | Mud |
| Otter trawl doors | 2 | 3 |
| Beam trawl shoes | 3 | 3 |
| Sweeps, chains and bridles | 1 | 2 |
| Tickler chains | 2 | 3 |
| Ground gear subsurface | 2 | 3 |
| Ground gear surface | 1 | 1 |

The daily SAR is calculated as the total SAR of all gear components:

$$\text{SAR}_{\text{daily}}^{k,l} = \sum_j \frac{T_{\text{trawl}}^{j,k} \cdot w_{j,l} \cdot v_j}{1 \text{ day} \cdot A_k},\qquad(4)$$

where $w$ is the total width of all gear components penetrating into layer $l$.

### 2.3. Model implementation of bottom trawling impacts

The daily trawling fields generated as described in 2.2 are interpolated to the unstructured model grid and implemented as additional forcing during the computation.

#### 2.3.1. Sediment resuspension

The daily trawling resuspension rate calculated according to Eq. ( 3 ) is added to the natural hydrodynamic resuspension rate at each model time step. The particle size distribution in the suspension wake of a trawl has been shown to be similar to that of the seabed surface (O'Neill and Summerbell, 2011). Therefore, the trawling resuspension rate is divided among the sediment classes according to their fractions in the seabed. The resuspended sediment is distributed evenly over the bottom layer of entire grid cell, where it can be mixed upwards by turbulence and advected horizontally to neighboring grid cells, or redeposited





in the absence of currents. This approach has been previously applied successfully in the Baltic Sea by
Porz et al. (2023).

### 2.3.2. Anthroturbation

The instantaneous mechanical action of penetrating gear can homogenize the sediment column down to
the penetration depth (Oberle et al., 2016b). Spiegel et al. (2023) attributed an exceptionally strong mixing
signal in a sediment sample retrieved from the Skagerrak to trawling. In a 1-D modeling study, de Borger
et al. (2021b) assumed total homogenization to account for physical trawling disturbance. However,
because the grid cell areas in our model are typically much larger than the daily swept area ($SAR_{daily} \ll$
1), the instantaneous mixing action cannot be resolved directly through homogenization, as this would
strongly overestimate mixing. Instead, anthroturbation is considered a diffusive process in this study,
analogous to bioturbation. This approach is similar to that implemented by Duplisea et al. (2001), who
accounted for physical trawl disturbance within a 0-D box model through a "physical mixing modifier",
effectively increasing the exchanges of OC between shallower and deeper compartments.

In order to find appropriate diffusion coefficients for each depth level of penetration, we consider that for
$SAR_{daily} = 1$, the sediment column should be well mixed down to the penetration depth after one day.
The diffusion coefficients are determined numerically by starting from a typical vertical OC profile and
finding the diffusion needed to reduce the vertical concentration gradient down to 10% after one day (see
Appendix C for details). The diffusion coefficients found for depth levels one to three are 2.42, 19.92,
and 86.51 cm$^2$ d$^{-1}$, respectively. During the simulation, these coefficients are scaled by the SAR, resulting
in an effective daily diffusion, which is applied to the corresponding penetration depth interval. The
average effective diffusion in trawled areas applied in this way for depth levels one to three are 0.020,
0.077 and 0.076 cm$^2$ d$^{-1}$, respectively. This is the same order of magnitude as expected natural
bioturbation intensities in the North Sea (Teal et al., 2008), but locally exceed expected bioturbation in
heavily trawled areas such as the Skagerrak (see Figure C2), in agreement with the observations by
Spiegel et al. (2023).



### 2.3.3. Macrobenthos depletion

Trawling is known to deplete seabed biota due to physical disturbance (Hiddink et al., 2017; Sciberras et al., 2018). In this study, a depletion rate of $d = 20\%$ per trawl pass is assumed, which corresponds to the mean reduction in benthic community abundance found by Sciberras et al. (2018) in a global meta-analysis. This depletion rate is scaled with the daily SAR at each depth level to generate the daily effective depletion, and the resulting biomass at time step $t$ is scaled accordingly:

$$B_t^l = B_{t-1} \cdot \left(1 - d \cdot \mathrm{SAR}_{\mathrm{daily},t}^l\right). \tag{5}$$

### 2.4. Management scenarios

In addition to the reference simulation (REF) using the actual trawling distribution, we simulate a scenario without any bottom trawling (NON) and four management scenarios of trawling redistribution in the North Sea relating to trawling closures in: 1. Marine Protected Areas (MPA), 2. Offshore Wind Farms (OWF), 3. Core Fishing Grounds (CFG), and 4. Carbon Protection Zones (CPZ).


#### 2.4.1. Marine Protected Areas

Though several areas in the North Sea have been designated as MPAs by national and international governmental entities, few restrictions on bottom trawling activity are currently implemented and enforced within them. In this scenario, we assume bottom trawling closures in all MPA polygons

contained in the World Database on Protected Areas (WDPA; UNEP-WCMC and IUCN, 2022), representing an extreme case of trawling closure enforcement within MPAs.

#### 2.4.2. Offshore Wind Farms

Vessel traffic in general, and use of bottom-contacting fishing gears especially, is usually restricted or

banned within OWFs. A substantial increase of conflict potential between OWFs and fisheries in the North Sea is therefore to be expected during the next decades, with several riparian nations' plans for the construction of extensive offshore renewable energy infrastructure (Stelzenmüller et al., 2022). For this scenario, wind farms in the 4C Offshore database (https://www.4coffshore.com/windfarms/, version of May 02, 2023) are considered, excluding those where the project status is classified as "Cancelled",





"Decommissioned", or "Failed Proposal". The scenario thus also includes projects that are at a developmental stage in addition to those that are operational or under construction, representing a maximum future development within the next decades. Some existing, nearshore OWFs are smaller in extent than the resolution of trawling effort (0.1°×0.1°) and have therefore not been considered.

Wind turbines change the regional hydrodynamic conditions through the generation of wind wakes,
effectively decreasing wind speeds downwind of the turbines (Akhtar et al., 2022). Additionally, the presence of turbine piles increase the hydrodynamic turbulence locally within the OWFs, with regional impacts on currents and stratification (Christiansen et al., 2023). We adopt the parametrizations of Christiansen et al. (2022a; 2022b; 2023), which were developed and validated for the North Sea, to account for the wind wake and pile effects. Detailed explanation and parameter settings of the OWF wind
wake and turbulence models are given in Appendix D. We additionally simulate different combinations of wake and pile effects and wind reduction for one year in order to gauge their relative importance for OC redistribution.

### 2.4.3. Core Fishing Grounds

ICES (2021) proposed a scenario which would restrict demersal fishing to core areas that are already heavily impacted with the goal of protecting habitat in the peripheral grounds. In this scenario, trawling is restricted to grid cells with the highest 90% of SAR based on the total mobile bottom contacting fishing gear intensity, averaged for the period 2013–2018. The resulting polygons were simplified to avoid a highly fragmented landscape that would be difficult to implement, communicate and enforce.


### 2.4.4. Carbon Protection Zones

In this scenario, areas where sediment OC is most vulnerable to disturbance are declared as Carbon Protection Zones (CPZs). We define the vulnerability $V$ as the maximum potential (oxic) carbon remineralization rate in the uppermost 10 cm sediment, calculated from the winter fields of the long-term
TOCMAIM simulations (Fig. 4):



$$V = (1 - \phi) \cdot \rho \cdot \Delta z \cdot \sum_{l,i=1}^{i=3} r_i \cdot OC_{i,l}, \qquad (6)$$

where $\Delta z$ is the layer thickness, $\phi$ is sediment porosity, $\rho = 2650 \ \text{kg m}^{-3}$ is the sediment grain density, $OC_{i,l}$ are the fractions of the three carbon pools in sediment layer $l$ and $r_i$ are their respective oxic remineralization rates $(r_1 = 5.5 \times 10^{-2} \ \text{d}^{-1}, r_2 = 5.5 \times 10^{-3} \ \text{d}^{-1}, \ r_3 = 5.5 \times 10^{-5} \ \text{d}^{-1})$. The vulnerability thereby takes into account both the total OC content and its lability to remineralization when resuspended or exposed to the sediment-water interface. We delimit CPZs as areas where the modeled potential OC remineralization rate exceeds 20 mmol m$^{-2}$ d$^{-1}$, corresponding to the top 20$^{\text{th}}$ percentile of values in the North Sea. In this way, the total amount of amount of redistributed trawling effort is similar to that of the CFG and MPA scenarios (Table 3).

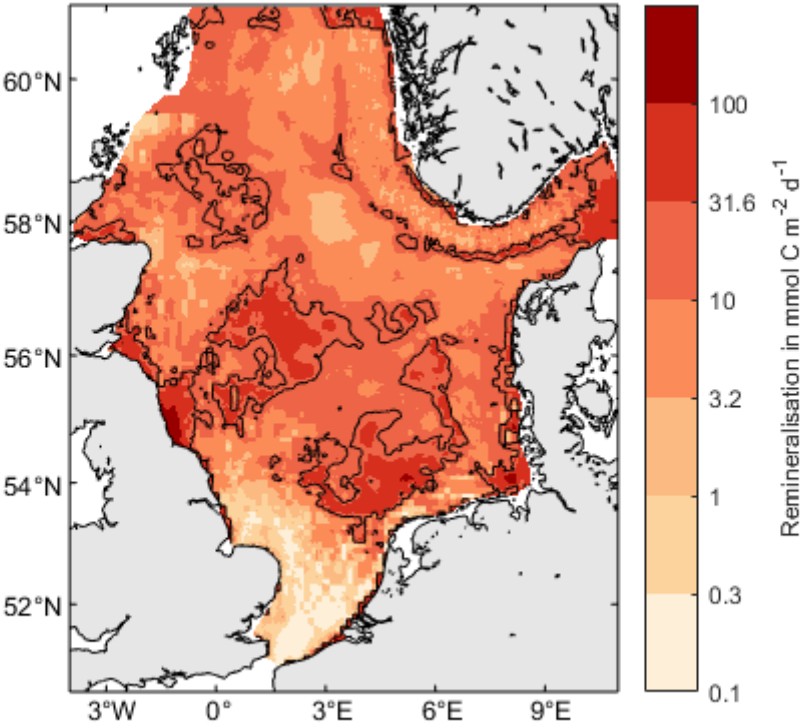

Figure 4. Carbon vulnerability. The map shows the maximum (oxic) carbon remineralization rate in the uppermost 10 cm sediment from the winter fields of OC content based on the long-term TOCMAIM simulations, taken at the end of the year 2013. Contour lines are shown at 20 mmol C m$^{-2}$ d$^{-1}$.





### 2.4.5. Redistribution of trawling effort

Both SAR and resuspension depend on sediment type and métier, and thereby on location. Therefore, a redistribution of those forcing fields alone is not sufficient for estimating seabed impacts following a redistribution of trawling effort. Instead, a redistribution of trawling effort itself is necessary.

For the relocation of trawling effort, we assume that

1) vessels will relocate to areas outside of the closure zones,

2) the total trawling effort (in terms of fishing hours) will remain unchanged,

3) the relative spatial and temporal distribution of trawling effort outside of the closure zones will remain similar, and

4) trawling effort will remain within approximately 1° of the previous fishing grounds.

The redistribution of trawling effort is implemented as a reallocation of daily trawling hours from vessels

inside to vessels outside of the closure areas, but within the limits of 1°×1° cells, and proportional to the existing effort:

$$f^k_{\text{scenario}} = f^k_0 \cdot \left( 1 + \frac{\sum_l f^l_0}{\sum_k f^k_0} \right),$$

$$f^l_{\text{scenario}} = 0,$$

( 7 )

where $f_0$ is the existing trawling effort before closure, and $k$ and $l$ indicate forcing grid cells outside and inside of the closure zones, respectively. Whenever a 1°×1° cell contains a closure zone with trawling effort, but no trawling effort outside of the closure zone, such that no effort can be redistributed within

the cell, the effort is distributed over the remaining domain instead, using the otherwise identical method of redistribution. Though this violates assumption (4), it is necessary to ensure that total trawling effort remains unchanged in accordance with assumption (2). The resulting trawling effort is processed as described in Sect. 2.2 to calculate daily resuspension and SAR fields. Figure 5 shows the changes in average trawling resuspension rates for the four management scenarios, with some statistics listed in Table

3.





Table 3. Simulated trawling management scenario statistics. The untrawled portion of OC refers to areas where annual surface SAR<0.01 for the averaged fields of 2015-2020 on a 0.1°x0.1° grid, taking into account OC content and porosity. All values are in percent.

| Scenario | Portion of North Sea closed | Redistributed trawling effort | Untrawled portion of sedimentary OC | Change in avg. trawl resuspension rate (Scenario−REF) |
|----------|------------|------------|------------|------------|
| REF | 0.0 | 0.0 | 12.84 | - |
| NON | 100.0 | - | 100.0 | −100.0 |
| MPA | 18.93 | 28.20 | 22.68 | +3.26 |
| OWF | 9.46 | 4.76 | 17.57 | −0.47 |
| CFG | 60.98 | 28.02 | 60.76 | +4.74 |
| CPZ | 23.23 | 28.81 | 42.02 | −11.03 |





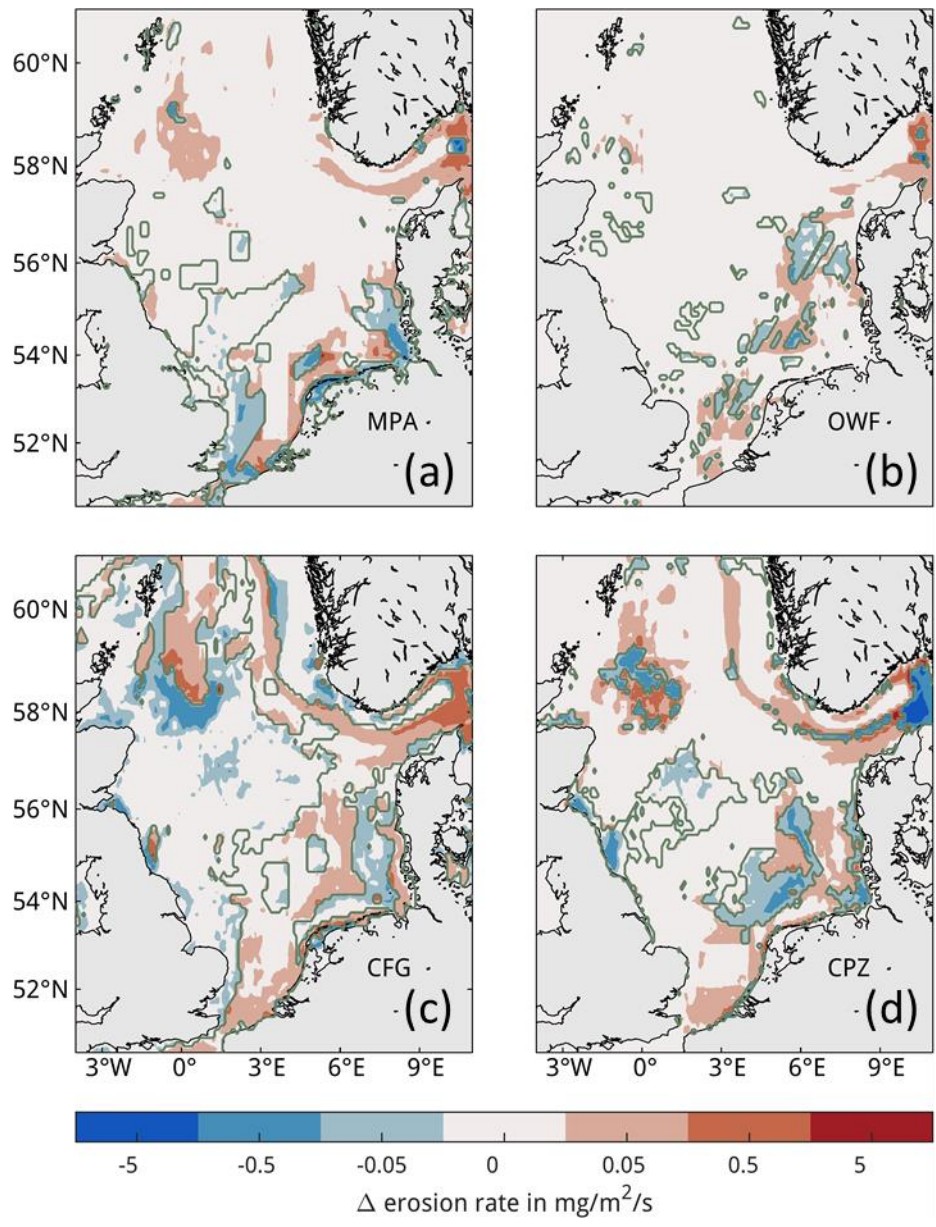

Figure 5. Redistribution of trawling for management scenario simulations. Colors show the change in average trawling resuspension rate after redistribution of trawling effort from inside to outside of the trawling closure zones (outlined) compared to the reference simulation with actual trawling effort, where positive values indicate an increase in resuspension rate following redistribution. Scenarios are (a) Marine Protected Areas, (b) Offshore Wind Farms, (c) Core Fishing Grounds and (d) Carbon Protection Zones.





## 3. Results

The overall impacts of trawling are examined by comparing the simulation results without trawling (NON) and the reference simulations (REF) with actual bottom trawling effort (Fig. 6a). The difference reveals that trawling causes both areas of OC loss and gain in the sediment. Loss of OC is highest in areas

that are both heavily trawled and rich in OC, namely the Skagerrak, the edge of the Norwegian Trench, Fladen Ground, Oyster Ground, the German Bight and part of the British coast. Gain of OC is seen in the peripheral areas of heavily trawled areas, most notably in the untrawled central Norwegian Trench. The spatial pattern of carbon gain and loss is relatively stable temporally, with only few patches showing inter-annual variations in gains and losses (Fig. 6b). The average end-of-year impact is a net loss of both OC

and macrobenthos biomass due to trawling, estimated to 552.2 kt of OC and 340.7 kt (ash-free dry weight), respectively (Table 4). This loss in biomass corresponds to about 14% of the total simulated macrobenthos biomass in the North Sea.

In the management scenarios, only the MPA and CPZ scenarios show an average increase in OC compared to the REF simulation (Table 4). The increase is moderate for the MPA scenario at only 5%,

while the increase in the CPZ scenario reaches nearly 30%. The OWF and CFG scenarios show a small negative change on carbon, on average. In the OWF scenario, impacts extend far beyond the OWFs due to the wake effects on hydrodynamics. The negative impact of trawling redistribution on OC is mostly mitigated by the reduced wind field inside of the OWFs due to reduced resuspension by natural currents (see F). Macrobenthos biomass is increased for all scenarios and all years compared to the reference

scenario, with greatest positive impacts in the CPZ and CFG scenarios, where net biomass increases by 53% and 28% relative to the REF simulation, respectively.

There is considerable seasonal variability in the OC impacts in the management scenarios, and to a lesser degree in the biomass impacts (Fig. 8). Especially in the CPZ and CFG scenarios, the increase in OC is reduced during the summer months when trawling pressure is highest. In the CFG scenario, the change

in OC compared to the REF simulation even becomes negative during summer. However, this effect essentially vanishes by the end of the year.

The net end-of-year OC changes in the scenarios show considerable inter-annual variability compared to REF and respond predictably to the level of trawling pressure (Fig. 9), rising until 2002 and declining



subsequently. Positive changes to OC tend to increase with increasing trawling pressure for all scenarios.
End-of-year changes in OC compared to REF are positive for all years in the NON, MPA and CPZ scenarios, whereas they become negative in some years for the OWF and CFG scenarios.

The general spatial pattern of changes in OC in the trawling redistribution scenarios is an increase within the closure zones and a decrease outside of the closure zones (Fig. 7). Locally, however, the opposite pattern also occurs both inside and outside of the closure zones. Some closure zones show little to no 380 changes in OC at all, such as the MPA covering the Dogger Bank (Fig. 7a).

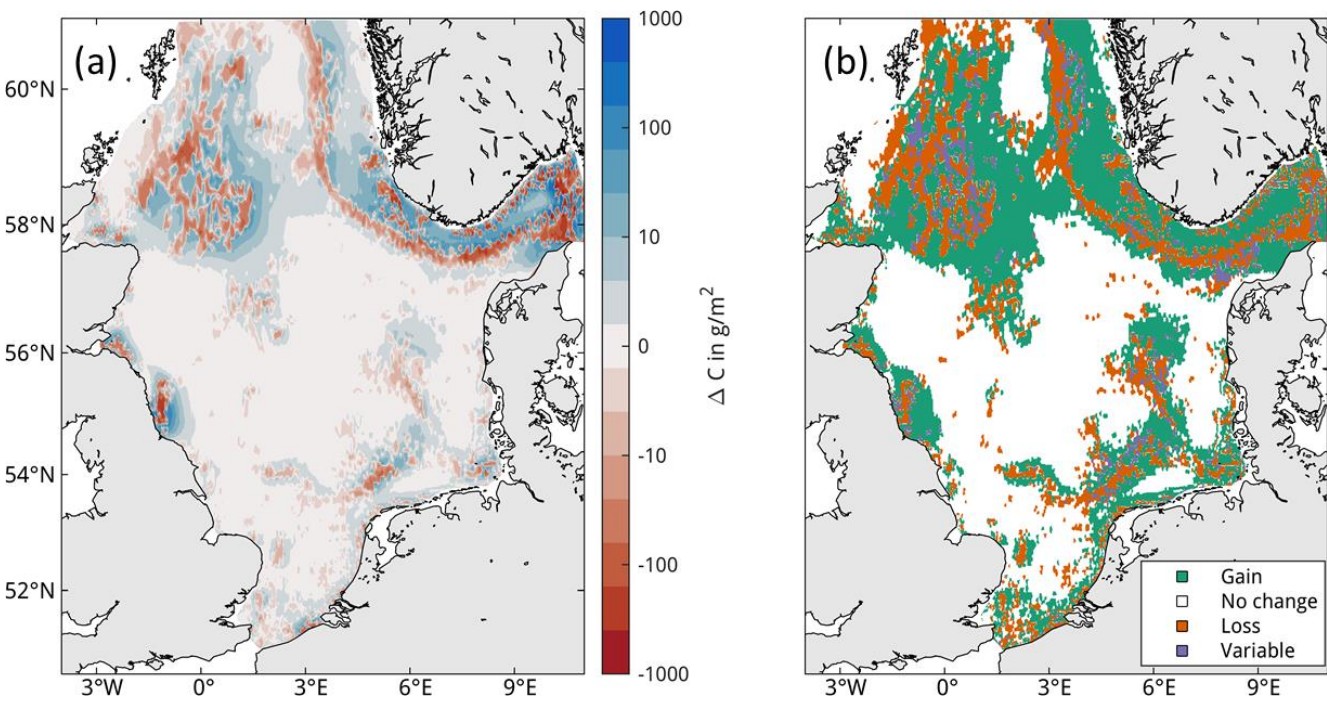

Figure 6. Spatial trawling impacts on OC. (a) Average end-of-year difference in total sediment OC between the no-trawling scenario and the reference simulation (NON−REF) for the years 2000-2005, where positive values indicate an increase in sediment OC due to trawling. (b) Inter-annual consistency 385 of changes. "Gain" and "Loss" indicate areas that show consistent OC gain and loss due to trawling at the end of each of the simulated years, respectively. "Variable" indicates areas in which both OC loss and gain occurred. "No Change" indicates areas in which maximum absolute OC changes do not exceed 1 g m$^{-3}$.





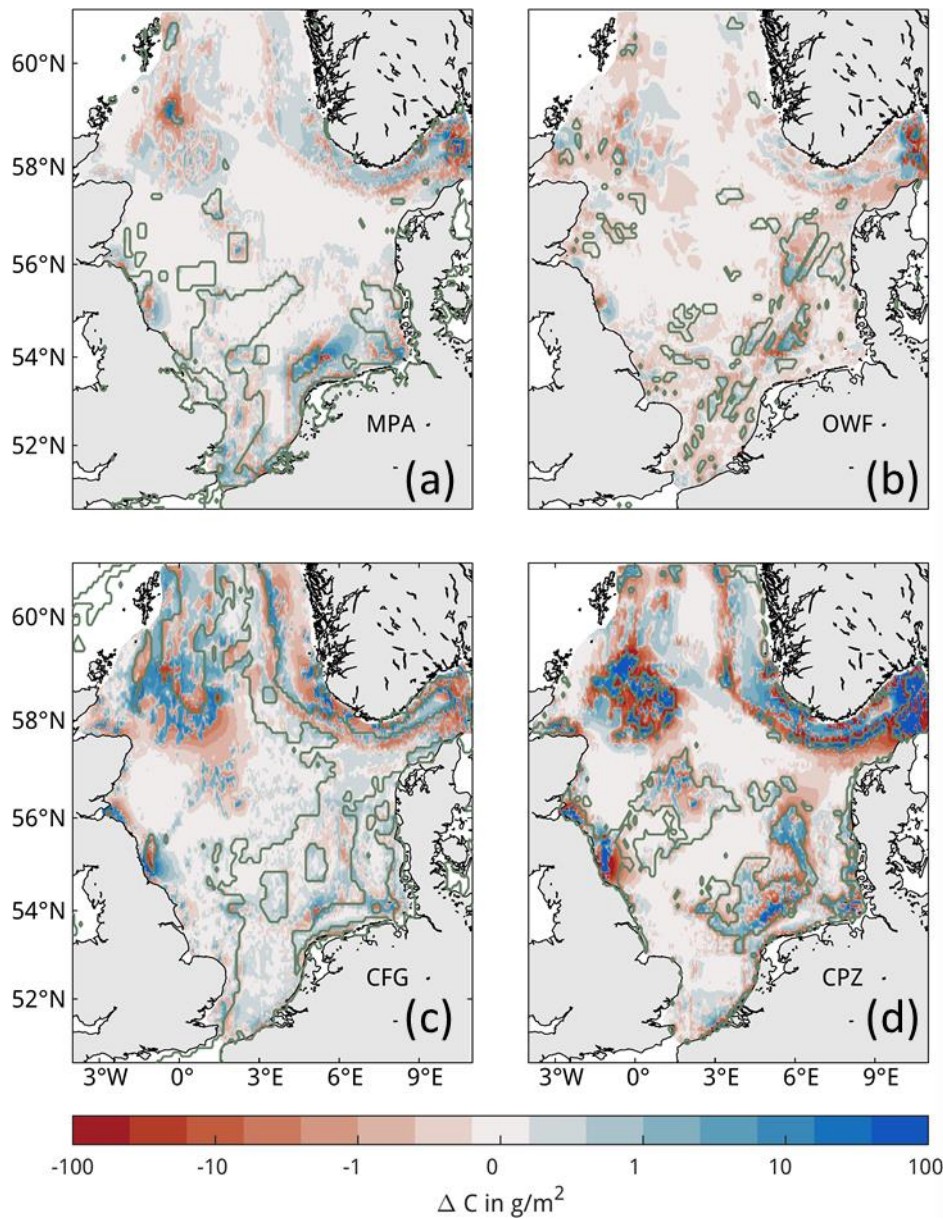

Figure 7. Spatial impacts in trawling management scenarios. Impacts are shown as average differences to the reference simulation with actual trawling effort at the end of each year, where positive values indicate an increase compared to the reference simulation. Scenarios are (a) Marine Protected Areas, (b) Offshore Wind Farms, (c) Core Fishing Grounds and (d) Carbon Protection Zones with respective closure zones outlined.





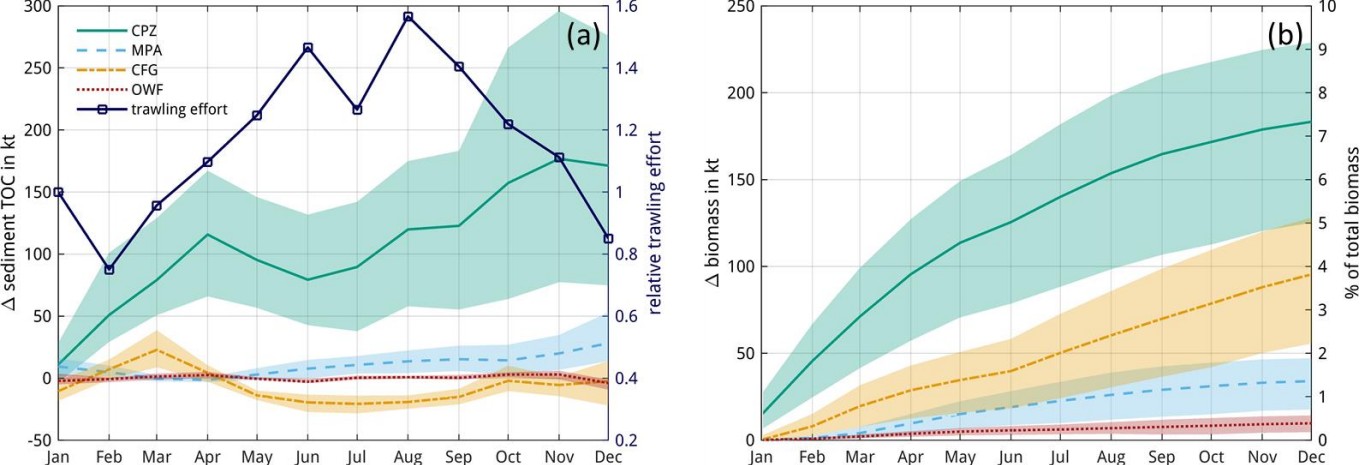

Figure 8. Seasonal impacts in trawling redistribution scenarios. Impacts are shown as differences to the reference simulation with actual trawling effort, where positive values indicate an increase compared to the reference simulation for (a) sediment organic carbon and (b) macrobenthos biomass. Lines and shading indicate monthly means and ranges of values for the years 2000-2005, respectively. The relative monthly trawling effort with respect to January is additionally shown in (a). Macrobenthos biomass is in ash-free dry weight. The right vertical axis in (b) approximates the differences as a proportion of the total macrobenthos biomass, taken here to be 2500 kt.

Table 4. Net impacts of trawling redistribution scenarios. Averages and standard deviations of the changes of OC and biomass are given as end-of-year differences to the reference simulation (Scenario−REF) for the years 2000−2005. Macrobenthos biomass is in ash-free dry weight. Positive values indicate an increase of mass compared to the reference scenario. Relative impact is calculated with respect to the scenario without trawling ([Scenario−REF]/[NON−REF]) for each year and then averaged.

| Scenario | Sediment OC impacts | | Macrobenthos biomass impacts | |
|---|---|---|---|---|
| | in kilotons | Relative to total impact (%) | in kilotons | Relative to total impact (%) |
| NON | +552.2±192.4 | +100 | +340.7±65.1 | +100 |
| MPA | +29.1±16.0 | +5.0±1.1 | +34.0±11.6 | +9.7±1.7 |
| OWF | −3.4±3.6 | −0.6±0.5 | +9.8±3.8 | +2.8±0.6 |
| CFG | −1.9±14.1 | −1.1±3.3 | +96.5±28.5 | +27.8±3.3 |
| CPZ | +167.1±78.5 | +29.2±5.2 | +183.6±40.9 | +53.6±1.9 |





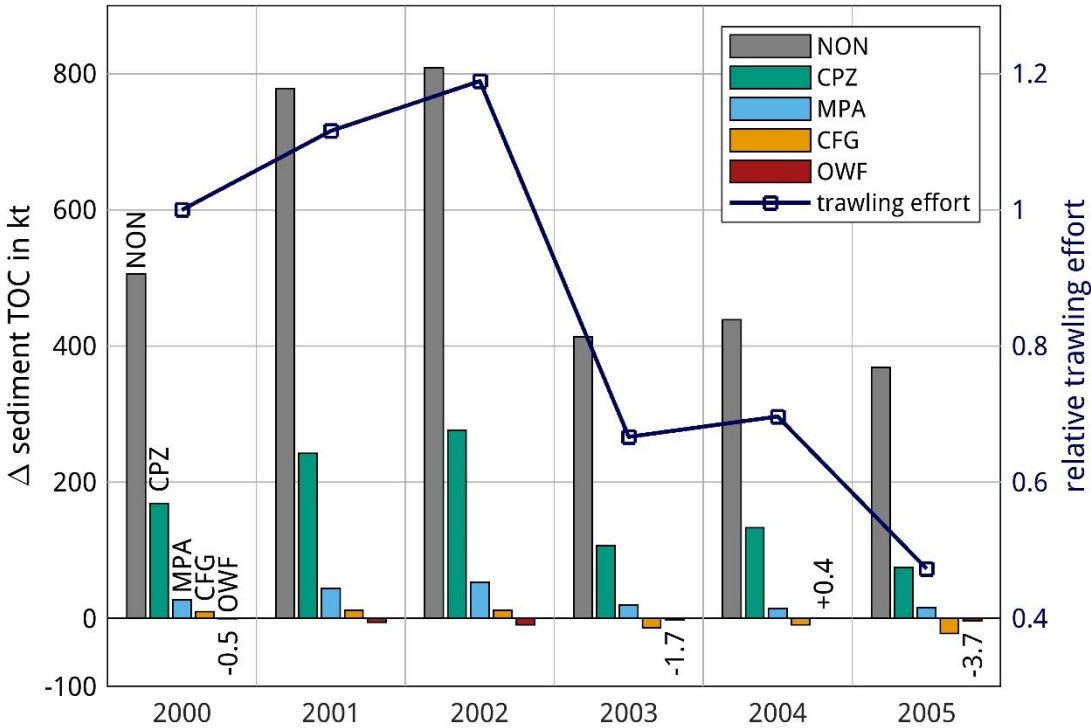

Figure 9. Annual impacts of trawling scenarios on sediment organic carbon. Impacts are shown as differences to the reference simulation with actual trawling effort at the end of each year, where positive values indicate an increase compared to the reference simulation. Individual bars are labeled with their numeric values for legibility. The relative annual trawling effort is shown with respect to the year 2000.

## 4. Discussion

### 4.1. Overall trawling impacts

The overall impact of bottom trawling on sedimentary OC is estimated in this study to the order of several hundred kilotons excess loss per year. This corresponds to a sizable portion of the annual sedimentation of OC in the North Sea, which has been estimated to the order of 1000 kt yr$^{-1}$ (de Haas et al., 1997; Diesing et al., 2021). The area found to be of greatest importance regarding trawling impacts to OC is the Skagerrak, where a combination of high trawling pressure and high mud and OC contents cause exceptionally high OC loss and redistribution. The considerable spatial variability of OC changes, with some untrawled areas adjacent to trawling grounds showing a gain in OC due to the transport and




redeposition of resuspended carbon, highlights the crucial role of lateral transport and redeposition of OC
resuspended by trawling.

The decline of macrobenthos biomass in trawled areas is mainly attributable to the excess mortality
induced by trawling, and partially to the removal of available food in the form of OC from the sediment
by trawling resuspension. In contrast to OC, biomass does not increase much in untrawled areas of
enhanced deposition (Figure E1). The reason is that highly labile fractions are remineralized during
transport in the water column, while most redeposited OC is refractory and cannot be utilized efficiently
by macrobenthos for growth in the model.

Trawling impacts respond linearly to the inter-annual changes in the level of trawling pressure, leading
to significant inter-annual variability in net OC impacts. This source of variability is starkly apparent in
our simulations during the strong decline of trawling pressure in the early 2000's.

The spatial patterns of trawling impacts are relatively stable inter-annually. This apparent stability may
be exaggerated in the model due to the use of a daily "climatology" of trawling impacts generated from
multiple years' trawling effort data, meaning that the spatial distribution of trawl forcing is identical in
each year, with only the intensity differing between the years. In reality, trawling effort does vary locally
from year to year, so the actual stability in trawling impacts on OC may be overestimated in our result.
Nevertheless, the results indicate that inter-annual hydrodynamic variability does not exert significant
control on OC redeposition patterns in the North Sea.

### 4.2. Impacts of management scenarios

Macrobenthos biomass has a straightforward spatial response to trawling closures in the management
scenarios (Figure E2), showing increase inside and decrease outside of the closure zones. The net impact
on total biomass is positive for all scenarios. This result aligns with the findings in ICES (2023a), who
argued that confining trawling pressure spatially strongly reduces impacts on benthic status at
comparatively small cost in terms of total catches or value. Changes to OC, however, show a more
complex response to trawling closures in our results.

Trawling closures in MPAs have a positive impact on OC storage in the simulations. It is notable that
carbon benefits occur in this scenario in spite of the higher average trawling resuspension rates. This can





be explained by the nonlinear impacts of trawling effort on OC: If trawling effort is redirected to trawled areas where OC has already been depleted, the additional effort will not cause much change in OC, whereas even moderate trawling effort can quickly deplete OC in previously undisturbed areas. The

MPAs showing little changes in OC feature low OC content, low trawling pressure, or both such as the MPA covering the Dogger Bank. Trawling closures in these areas will therefore have limited impact on the OC budget.

The OWF scenario shows a very small impact on overall OC and biomass, owing to the limited overlap between trawling grounds, OWFs and areas of OC deposition as well as the mitigating factor of reduced

wind speed, which decreases wind stress and thereby natural current resuspension within and downwind of OWFs. Strong local effects do occur, and the wind wakes and pile turbulence production cause the impacts to extend to considerable distances from the OWFs themselves. A holistic assessment of OWF impacts on sediment OC should consider the OC loss due to seafloor disturbance during construction and decommissioning, as well as secondary effects, such as the colonization of organisms at the foundations

of wind turbines, and wind wake impacts on the ecosystem structure (de Borger et al., 2021a; Daewel et al., 2022; Heinatz and Scheffold, 2023). Using an ecosystem model that considers wind wake effects, Daewel et al. (2022) simulated local increases in sedimentary carbon of up to 10% after one year, but only a slight net increase of 0.2% for the entire North Sea. Though our OWF scenario shows a slight decrease in OC, this is primarily due to the trawling effort redistribution, whereas the wind wake effect

shows a similar sign and magnitude as in Daewel et al. (2022). Heinatz and Scheffold (2023) estimated a net increase of sediment OC storage at OWFs in the Southern North Sea on the order of 1000 kt throughout a 20 year life cycle, or 50 kt yr$^{-1}$. This estimate is an order of magnitude greater than our estimated reduction. Overall, a comparison of these budgets implies a net positive impact of OWFs on OC storage, with local impacts at the foundations outweighing the large-scale impacts that redistribute

OC in the far field. Plans for areas of OWF development have expanded rapidly during recent years and continue to evolve quickly, making an increased impact on OC in the future likely.

The CFG scenario, which is aimed at habitat preservation, shows no significant impact on OC, and a moderate positive effect on biomass. Considering that this scenario requires by far the largest portion of





the North Sea (>60%) to be closed to trawling, this management option can characterized as rather low
in efficiency.

The highest carbon and biomass protection is achieved with the CPZ scenario, which is specifically aimed
at sediment carbon protection and is almost six times more effective than the MPA scenario in terms of
OC preservation and almost twice as effective as the CFG scenario in terms of macrobenthos biomass
preservation at comparable trawling effort displacement.

### 4.3. Implications for climate impacts

The average sediment carbon loss from trawling predicted in this study is equivalent to subaqueous
emissions of 2.0 Mt $CO_2$ $yr^{-1}$, or 4.0 t $CO_2$ $km^{-2}yr^{-1}$ averaged over the trawled area. This is significantly
lower than the 118 t $CO_2$ $km^{-2}yr^{-1}$ computed by Sala et al. (2021) for trawled areas globally, even though
the impacts in the heavily trawled North Sea can be expected to be higher than the global average.

The ultimate climate impacts of these emissions generally depend on the efficiency with which $CO_2$ in
the bottom waters is mixed towards the surface layers and exchanged with the atmosphere, which is
controlled by water depth and regional circulation patterns (Collins et al., 2023). In the North Sea,
ventilation of the shallower and partially mixed Southern Bight can be expected much faster than for the
deeper and stratified Norwegian Trench. Atwood et al. (2024) estimated that globally, 55–60% of
trawling-induced aqueous $CO_2$ emissions accumulate in the atmosphere within the decades following
trawling, which would place the climate impacts in our results to the order of 1 Mt $CO_2$ $yr^{-1}$.

It is worthwhile to distinguish between the concepts of carbon storage and carbon sequestration in the
context of climatic impacts; the former refers to the total amount of carbon contained the sediment,
whereas the latter refers to the rate of removal from the carbon cycle through continuous sedimentation.
From the perspective of climate change mitigation, sequestration can be seen as the more important factor,
since it represents a continuous carbon sink in the Earth system. In the North Sea, several areas of active
carbon sequestration are among the most intensely trawled, most notably the Skagerrak (Diesing et
al., 2021).



### 4.4. Implications for marine spatial management

Smeaton and Austin (2022) and Black et al. (2022) have argued in favor of prioritizing muddy, inshore areas such as fjords and estuaries with high proportions of labile OC as carbon protection areas, whereas offshore sediments generally contain a lower portion of labile OC and are therefore less vulnerable to degradation. This is partially mirrored in our results, where the most OC-rich depocenter in the Norwegian Trench has a relatively low vulnerability (Fig. 4). However, some offshore areas also show significant

amounts of labile and semi-labile OC, designating them as priority areas for carbon protection as well. Indeed, Smeaton and Austin (2022) acknowledge that about 20% of offshore muddy OC is labile, which is not much less than in the inshore areas in their results (~30%). In addition, Graves et al. (2022) noted that the most labile OC will be remineralized regardless of human disturbance, and therefore the fractions of intermediate lability should be of most concern for management. We conclude that vulnerable offshore

OC deposits should be treated as equally important as inshore areas in the context of carbon management.

### 4.5. Model limitations

Trawling has been ongoing in the North Sea for many decades, yet our simulations only cover individual years, each initialized using the results of a longer-term sediment OC model (Zhang et al., 2019) in which trawling impacts were not considered. We nevertheless consider our results indicative of longer-term

impacts. The general spatial pattern of OC deposition is governed by ecosystem production and hydrodynamics and should therefore remain stable regardless of trawling activity. This deposition pattern is accounted for through the model's initial conditions, which represent a multi-decadal equilibrium of OC pools of different labilities. No additional OC deposition in the form of detritus following the phytoplankton blooms in spring and summer is added throughout the year, since the vast majority of this

seasonal detritus deposition is remineralized within the same year even without resuspension by trawling, with only the less labile portions remaining in the system for extended periods. Any further addition of fresh OC to the sediment surface is therefore not expected to change the findings significantly, though this shall be confirmed by longer-term simulations.

While we consider the carbon model used in this study fit for purpose, it is relatively simple and does not

include some of the less understood mechanisms that could modify the remineralization process. For



example, it has been shown that refractory OC can be made more degradable when mixed with labile OC, a phenomenon termed "priming" (Bianchi, 2011). It has been estimated that the priming effect increases the mineralization rate of refractory OC by 54%, on average (Sanches et al., 2021), though there is considerable uncertainty in this estimate and the mechanisms underlying priming are not well understood.

The homogenization of sediment during trawling could thereby increase the mineralization rate of the mixture, increasing the net loss of OC from the sediment, as observed by Paradis et al. (2019) and van de Velde et al. (2018). In addition, while this study focuses on remineralized OC, many shelf sediments also contain reduced inorganic species such as Fe(II) and sulfide (van Dam et al., 2022). When resuspended by trawls, these reduced compounds may quickly re-oxidize, releasing additional $CO_2$. These

considerations could enhance the impacts of trawling on direct aqueous $CO_2$ emissions compared to our estimate.

The dataset of fishing activity (Kroodsma et al., 2018) used in this study to generate trawl impact forcing fields is based mainly on Automatic Identification System (AIS) data, which does not include vessels smaller than 15 m length. Smaller vessels operate mainly in nearshore areas such as the Wadden Sea or

fjords, which are not resolved well in our hydrodynamic model. Impacts of small-scale, nearshore fisheries are therefore not represented in our results. Though it can be expected that smaller vessels have smaller impacts on the seafloor, some nearshore areas such as fjords and tidal flats are also hotspots of OC accumulation (Black et al., 2022), which will require high-resolution regional models to resolve properly.

Aside from bottom trawling, there are other bottom-contacting gear types, namely demersal seines and dredges, which have not been considered in this study. The ropes of demersal seines are expected to have a low hydrodynamic drag per unit length and no subsurface impacts. However, because the seine ropes can be several km in length, their overall impact may still be locally significant. Dredges are expected to have a high impact per surface area contacted (O'Neill et al., 2013), but they are utilized mostly off the

British coast (ICES, 2019) and do not interact directly with carbon depocenters.

Predicting a fleet's reaction to trawling closures is a complex problem involving political, societal and economic parameters that are outside the scope of this study. Our redistribution of trawling effort relies on the straightforward assumption that total trawling hours will not change due to closures. Püts et al.



(2023) used a trophic model for calculating redistribution which considers biomass, catchability, and
profitability of fishing the target groups. Such a method of redistribution may be more suitable for gauging
the tradeoffs between profitability and seabed impacts. Spillover effects causing an increase in effort
around the borders of the closure areas in their model, bearing some resemblance to our pattern of
redistribution.

We assume a constant benthos depletion rate per trawl pass of 20%, but benthos vulnerability is known
to be strongly dependent on species, gear type and sediment type (Hiddink et al., 2017; Sciberras et
al., 2018). Dependency on gear and sediment types are considered in our model, since the SAR fields are
resolved vertically. Consequently, deeper penetrating gears cause a stronger benthos depletion. In
contrast, species-dependency is not considered in our model. Long-term sensitivity to trawling impact has
been shown to generally increase with species longevity, because slower-growing benthic communities
take a longer time to recover following depletion (Rijnsdorp et al., 2018; Pitcher et al., 2022). As longer-
lived species tend to inhabit coarser-grained environments, our simulations may overestimate depletion
in muddy habitats and underestimate depletion in sandy and gravelly habitats. Such dependency may be
considered in future modeling studies, e.g. by prescribing spatially variable macrobenthos growth rates
in accordance with sampling data.

Finally, ecosystem feedbacks such as enhanced primary production through resuspension of nutrients, a
changing light climate caused by increased turbidity, or changes in alkalinity can affect carbon turnover
rates in shelf sea ecosystems (see Table 1). Resolving these feedbacks is necessary in order to get the full
picture of trawling impacts on the carbon cycle, which will require a two-way coupling between sediment
and ecosystem models.

**5. Conclusions**

We used a numerical model to quantify the impacts of bottom trawling on organic carbon and
macrobenthos in the sediment of the North Sea. We generated daily time series of resuspension and seabed
area contacted at different penetration depths on a 0.1°×0.1° grid using available information on vessel
activity, impacts of individual gear components and sediment type. These were used to force a coupled



3D numerical ocean, sediment and macrobenthos model for six consecutive years with different levels of

trawling pressure. The results were compared to reference simulations without trawl forcing.

In total, North Sea sediments contained 552.2±192.4 kt less organic carbon in the trawled simulations

than in the reference simulation by the end of each year, equivalent to aqueous emission of 2.0±0.7 Mt

$CO_2$, half of which is likely to accumulate in the atmosphere over multi-decadal timescales. The impacts

were elevated in years with higher levels of trawling pressure and vice versa. Significant spatial variability

in carbon redistribution was revealed, with some areas showing a strong loss of organic carbon due to

trawling, while nearby areas received more organic carbon following transport and redeposition. The area

most strongly impacted was the heavily trawled and carbon-rich Skagerrak. Average carbon loss per unit

area was smaller than a previous global estimate by a factor of nearly thirty (cf. Sala et al., 2021),

highlighting the need for detailed regional assessments in order to obtain the most accurate estimates of

trawling impacts.

Trawling reduced macrobenthos biomass by 340.7±65.1 kt in the model, corresponding to about

13.6±2.6% of total macrobenthos biomass present in the North Sea. These results underline the notion

that bottom trawling, analogous to fishing in general, must be understood as an integral component of the

North Sea ecosystem and carbon cycle, rather than a deviation from some intangible (and largely

unknown) "natural" state devoid of human influence.

We simulated four management scenarios, in which trawling effort was removed from potential trawling

closure areas and redistributed to nearby areas. Closures in planned Offshore Wind Farms and in Core

Fishing Grounds had negligible effects on net organic carbon, while closures in Marine Protected Areas

had a moderate positive impact. The largest positive impact emerged for closures in Carbon Protection

Zones, which were defined as areas where organic carbon is both reactive and abundant, and thus

particularly vulnerable to disturbance. In that scenario, the net impacts of trawling on organic carbon were

reduced by 29% and on macrobenthos biomass by 54%, demonstrating that carbon protection and habitat

protection may be combined through careful design of protection areas.

We consider this study to represent the most robust regional estimate of trawling impacts on sedimentary

carbon to date. Nevertheless, uncertainties remain regarding the validity of the results on longer

timescales, which will require long-term simulations to address. Further, the model can be improved by



adding additional representations of biogeochemical processes, such as food web feedbacks or the modification of benthic alkalinity fluxes by trawling, possible effects for which empirical data is lacking

thus far.




## Appendix

### A. Sediment model parameter settings

**Table A1. Sediment parameter settings.** For each fraction, set values are given for settling velocity
($w_s$), critical shear stress for resuspension ($\tau_c$), erosion rate ($M_E$; erosion formulation according to
Winterwerp et al. (2012)) and remineralization rate ($r$).

| | Inorganic | | | Organic | | |
|---|---|---|---|---|---|---|
| | Clay | Silt | Sand | Labile | Semi-labile | Refractory |
| $w_s$ (mm s$^{-1}$) | 0.005 | 0.1 | 0.2 | | 0.1 | |
| $\tau_c$ (Pa) | 0.1 | 0.1 | 0.2 | | 0.1 | |
| $M_E$ (s m$^{-1}$) | 0.001 | 0.001 | 0.002 | | 0.001 | |
| $r$ (d$^{-1}$) | - | - | - | $5.5 \cdot 10^{-2}$ | $5.5 \cdot 10^{-3}$ | $5.5 \cdot 10^{-5}$ |

### B. Hydrodynamic drag of gear components

Estimates for $H_d$ from Rijnsdorp et al. (2021) are used for nets, ground gear, shoes and tickler chains of
beam trawls and from O'Neill and Summerbell (2016) for the otter trawl doors (details in **Table B1**). The
values for ground gear and nets are assumed equal for beam and otter trawls. For ropes, sweeps and bridles
a value of 1 N m$^{-1}$ is used in accordance with O'Neill and Noack (2021).




**Table B1. Values of hydrodynamic drag of gear components used in the calculation of resuspension rates.** "Small" and "large" vessels refer to vessels with engine power smaller and greater than 221 kW, respectively.

| Gear component | Hydrodynamic drag (kN m$^{-1}$) | | Reference |
| --- | --- | --- | --- |
| | Small vessels | Large vessels | |
| Tickler chains | 0.699 | 2.118 | |
| Shoes | 0.019 | 0.013 | |
| Ground gear | 0.135 | 0.572 | Rijnsdorp et al. (2021) |
| Nets | 1.595 | 1.967 | |
| Otter doors | 1.0 | | O'Neill and Summerbell (2016) |
| Ropes, sweeps and bridles | 0.001 | | O'Neill and Noack (2021) |

## C. Determination of trawl mixing coefficients

In order to determine appropriate diffusion coefficients to describe the churning action of penetrating gear components, we start from a typical sediment OC profile, approximated using data in Zhang and Wirtz (2017) as

$$OC(z) = 10 + 20\,e^{-55z}, \tag{C1}$$

where OC is in mg g$^{-1}$ and $z$ is the sediment depth in meters (positive downward). A diffusion is then applied numerically to these profiles for maximum depths of 2, 5 and 10 cm, where the surface value is kept constant and a zero-gradient boundary condition is applied at the bottom. Diffusion coefficients are then found iteratively for each maximum depth such that after one day, the difference between surface and bottom values reaches 10% (Figure C1).



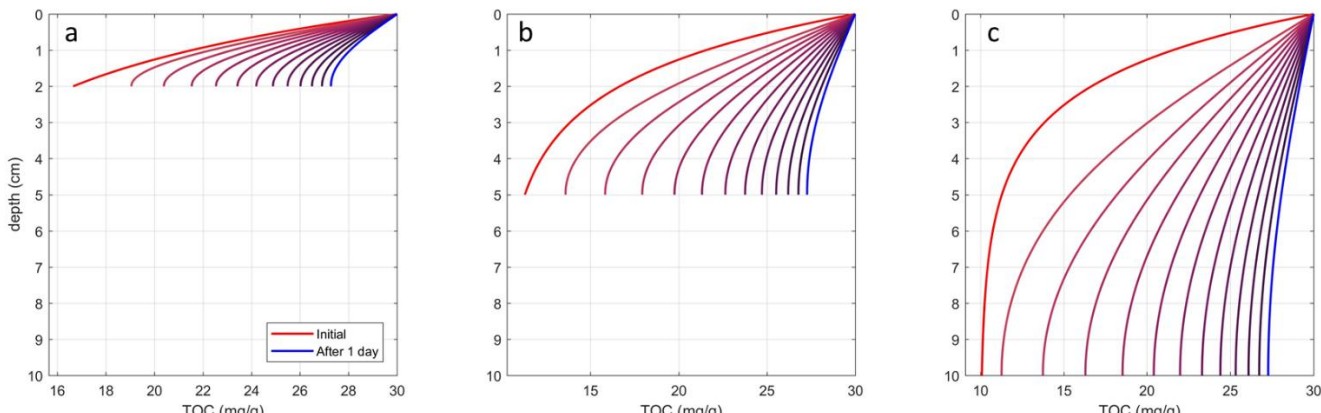

**Figure C1. Trawl diffusion coefficients.** The panels show the temporal evolution of an OC depth profile during one day at intervals of 2 hours for maximum depths of (a) 2 cm, (b) 5 cm, and (c) 10 cm when applying their determined respective diffusion coefficients. Initial profiles are the same for each case.

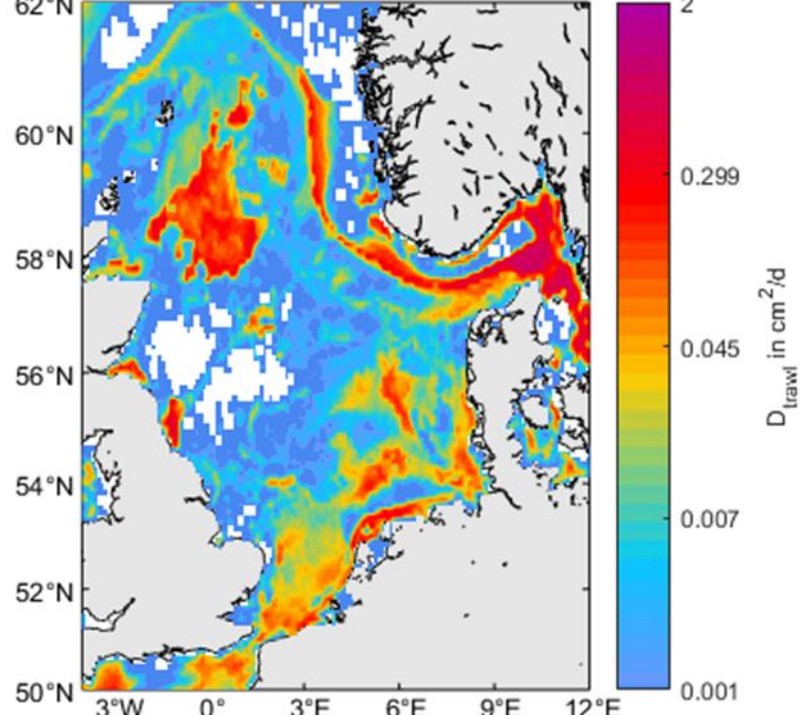

**Figure C2. Trawl diffusion coefficients.** Colors show the annually averaged diffusion coefficients applied to depth level 2 (2 –5 cm).



### D. Offshore Wind Farm parameterizations

In order to assess the impact of offshore wind farms on atmospheric and oceanic dynamics, additional parameterizations have been incorporated in the numerical equations of the SCHISM model. Here, we account for the reduction in surface wind speed within and behind offshore wind farms, as well as for the underwater drag and turbulence arising from offshore wind turbine foundations.

To quantify the reduction in wind speed, we adopt a top-down methodology proposed by Christiansen et
al. (2022a), which uses an empirical wake parameterization based on satellite observations to describe the deficit in surface wind speed $u_r$ in downstream direction. The 10-m wind speed $u_0$ is altered by an exponential function given as

$$u_r(x, y) = u_0 \left(1 - \alpha \cdot e^{-\left(\frac{x}{\sigma} + \frac{y^2}{\gamma^2}\right)}\right). \tag{D2}$$

Here, $x$ and $y$ denote the longitudinal and cross directions in the reference coordinate system, which is aligned with the respective wind direction, and $\gamma$ is a decay-constant related to the characteristic width of
the wind farm. Further details on this parameterization and its implications for ocean dynamics are elaborated in Christiansen et al. (2022a; 2022b). Our implementation utilizes a prescribed reduction in wind speed ($\alpha = 8\%$) and a constant wake length ($\sigma = 30\,\text{km}$), consistent with prior studies. Additionally, the parameterization is extended by a reduction of wind speed by $\alpha$ within the wind farm polygons, which is in line with atmospheric modeling of offshore wind wakes (Akhtar et al., 2022).

For addressing drag and turbulence induced by the underwater foundations of wind turbines, we adopt the sub-grid scale parameterization approach outlined by Christiansen et al. (2023), assuming that offshore wind turbines are built on monopile foundations. In this method, the drag force that a vertical cylinder exerts on an unstratified horizontal flow is considered via the model equations and can be expressed as

$$\vec{F}_d = -\frac{1}{2}\rho_0 C_d A_c |\vec{u}|\vec{u}. \tag{D3}$$

Here, $\rho_0$ is the density of the fluid, $C_d$ is the drag coefficient, $A_c$ is the frontal area of the cylinder that is exposed to the free stream, and $\vec{u}$ is the velocity of the free stream. For a comprehensive understanding of this parameterization and its performance, refer to Christiansen et al. (2023). For our implementation we adopt the model parameters given in the former study, using fixed structure properties with a drag



coefficient $C_d$ = 0.63 and a pile diameter of 8 m, and a pile distance of 1 km.
Exemplary impacts on physical fields are shown in Figure D1.

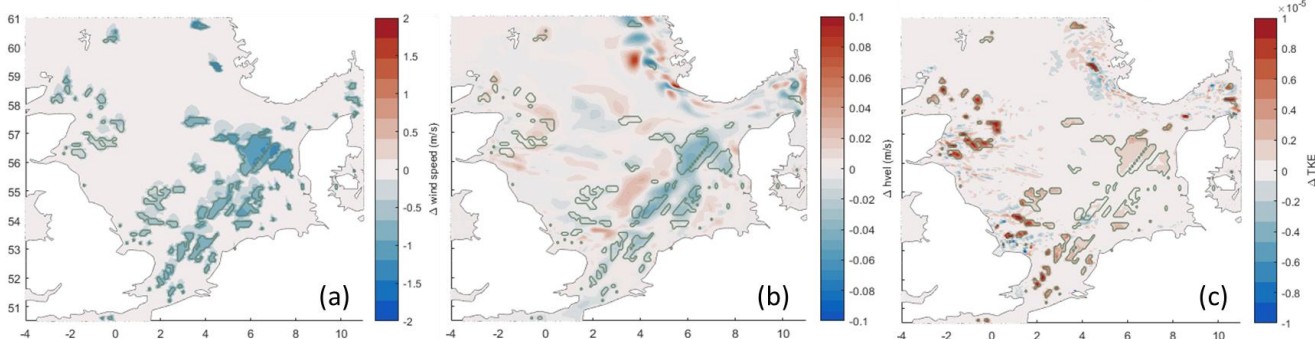

**Figure D1. Impacts of OWFs on physical fields.** Colors show the differences between the OWF scenario and the reference simulation averaged from hourly outputs on 31 December 2000: (a) 10 m wind speed, (b) horizontal near-surface current velocity, and (c) near-surface turbulent kinetic energy.




## E. Trawling impacts on macrobenthos biomass

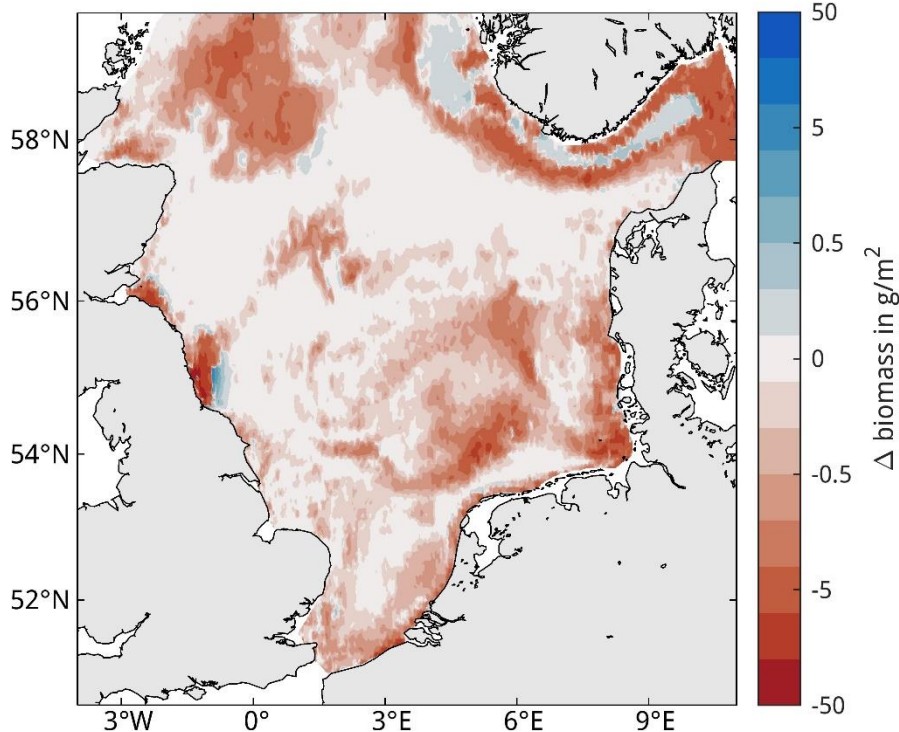

**Figure E1. Spatial trawling impacts on macrobenthos biomass.** Colors show the average end-of-year difference in total sediment OC between the no-trawling scenario and the reference simulation (NON−REF) for the years 2000−2005, where positive values indicate an increase in biomass due to trawling.




**Figure E2. Spatial impacts of trawling redistribution scenarios on biomass.** Impacts are shown as average differences to the reference simulation with actual trawling effort at the end of each year, where positive values indicate an increase compared to the reference simulation. Scenarios are (a) Marine Protected Areas, (b) Offshore Wind Farms, (c) Core Fishing Grounds and (d) Carbon Protection Zones with respective closure zones outlined in green.



### F. Offshore Wind Farm wake and pile impacts

Without wake or pile effects, net sediment OC is reduced by 2.3 kt compared to the reference simulation at the end of the year 2000. Including wake effects downwind of the OWFs and pile turbulence enhances the impact to a reduction of 2.5 kt. When wind reduction within the OWF is considered, the effect decreases to a reduction of 0.2 kt (Figure F1).

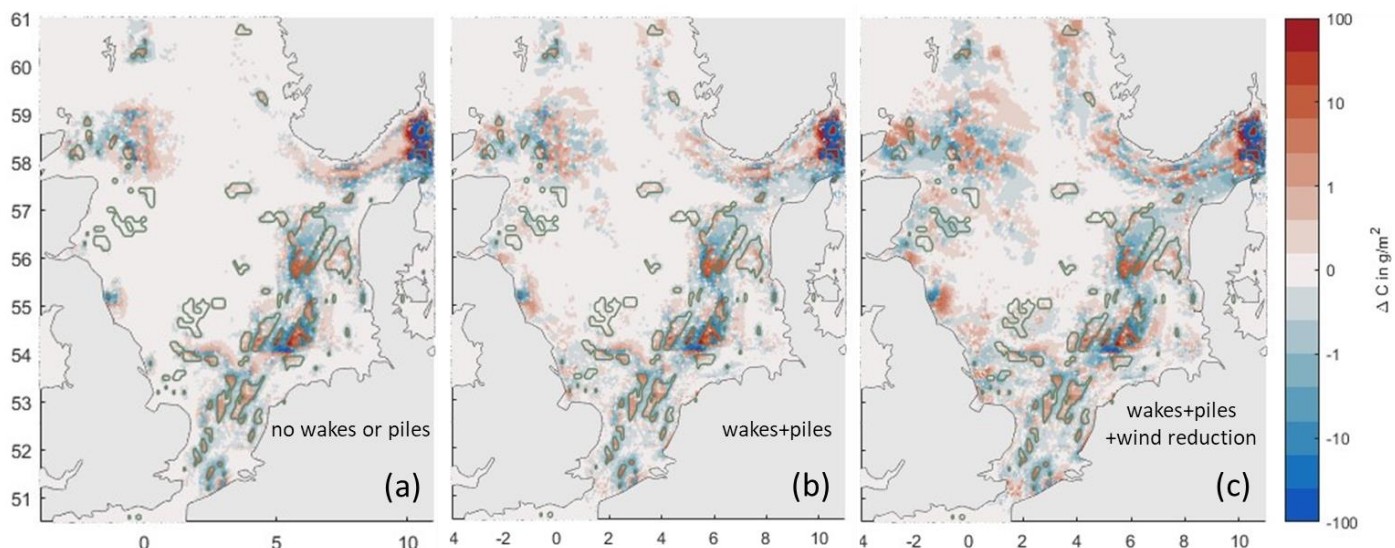


**Figure F1. Impacts of wake and pile effects in offshore wind farms on OC redistribution.** Colors show the average end-of-year difference in total sediment OC between the OWF scenario and the reference simulation for the year 2000: (a) Shows impacts due to trawling redistribution alone, (b) additionally includes wind speed reduction in the wakes of and enhanced turbulence inside OWFs, and
(c) additionally includes wind speed reduction inside of the OWFs.

**Code availability**

The SCHISM model including the sediment module is available at https://github.com/schism-dev. The TOCMAIM code is available at https://data.mendeley.com/datasets/2vvny3xd85/2/files/e68b1bb8-f1bc-4a6a-8f02-b149619f2c05. Code modifications done for the purpose of this study are available upon
request.

**Data availability**

**Author Contribution**

LP and WZ conceptualized the study. WZ and CS developed the project proposals leading to the study. LP designed the numerical model experiments and processed the outputs. NC implemented the wind farm
parameterizations. JK developed the hydrodynamic numerical model setup. WZ and UD provided initial fields for the TOCMAIM model. LP drafted the manuscript in consultation with all co-authors.

**Competing interest**

The authors declare that they have no conflict of interest.

**Acknowledgments**

This study is a contribution to the project "Anthropogenic impacts on particulate organic carbon cycling in the North Sea (APOC)" funded by the German Federal Ministry of Education and Research (BMBF) within the MARE:N program under grant 03F0874C. It is also supported by the Helmholtz research program POF IV "The Changing Earth – Sustaining our Future" within "Topic 4: Coastal zones at a time of global change". NC and JK are supported by the Cluster of Excellence EXC 2037 'Climate, Climatic
Change, and Society (CLICCS)' (Project Number: 390683824) funded by the German Research Foundation (DFG). NC is additionally supported by the CLICCS-HGF networking project funded by the Helmholtz Association of German Research Centers (HGF). This work used resources of the German



Climate Computing Centre (DKRZ) granted by its Scientific Steering Committee (WLA) under project ID bg1244.

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
