# Peer review of "Quantification and mitigation of bottom trawling impacts on sedimentary organic carbon stocks in the North Sea"

_EGUsphere, 2024_

## Author Comment (AC1)

**Reply to RC1** *(Responses in italics)*

**General comments**

Porz et al., conducted a modelling study which uses coupled 3D hydrodynamic simulations with a bioturbation model to look at trawling impacts in the North Sea. I think this may be the most holistic and up to date research on this topic that I've seen. Most studies on this topic have just looked at the direct impact of trawling on sediment carbon within a trawl track and some look at how that effect might evolve through time. This study looks at the potential fate of OC once resuspended and shows how trawling can lead to lower OC in some areas and more OC in other areas (something observed in field observations before as well). Furthermore, I was happy to see that in the simulations regarding fisheries closures, trawling effort was redistributed rather than assumed to have decreased. All the praise aside, I think the manuscript still has areas of improvement, most of which I've outlined below. My main concern is that, although, I've seen it mentioned throughout the paper, it was never clear to me exactly how remineralization was addressed by the model. For me, the parts about the removal of OC through resuspension and redistribution to other areas was better explained but I would like to know exactly in the methods, your assumptions on benthic mineralization after trawling as well as once resuspended. It is stated in the discussion that OC is degraded once resuspended but I would like to see further details behind this like how much mineralization decreased/increased (both in the water column and in the sediment) and why.

> *Thank you for the constructive comments that have helped us to clarify and improve our manuscript.*
>
> *We will extend the methods section to include more details on the model description. The degradation rate of each OC pool decreases with sediment depth to reflect the decrease in oxygen availability, and the maximum (oxic) remineralization rate is applied at the uppermost sediment layer (i.e. at the sediment-water interface) as well as to the OC suspended in the water column. We will adapt the methods section 2.1 to explain more clearly how OC degradation is implemented in the model. In addition, we will explain the mechanisms behind trawling resuspension on OC in the model more clearly in the results section 4.1.*

**Detailed comments**

L11 There are reasons why 3D hydrodynamic models provide much needed insight on this topic but this is not yet clear to the reader. It would be nice to transition from the debate amongst scientists to something like "However, current discussions around the fate of resuspended organic matter are lacking. To help resolve this, we used 3D hydrodynamic… "

> *Good point, this will be added to the abstract.*

L25 is there anything that could be said about what makes these sediments vulnerable? High labile OC/biomass, low natural disturbance etc.?

> *Yes, we defined vulnerability in terms of OC lability in the study, so this will be added to the abstract.*

L30 so does this give reason to advocate for spatial as opposed to effort management?

*In the context of carbon and habitat protection, our results may be used to argue in this direction. However, this is only one consideration for fisheries management, which has mostly prioritized prevention of overfishing. In addition, our results do not account for possible changes in catches following spatial effort redistribution. To prevent misunderstandings and considering the other reviewer's caution not to overstate our results, we prefer not to advocate for specific management strategies here.*

L49 "seabed destruction" is vague and perhaps too loaded of a term here. Can you be more specific?

*Rephrased to "Efforts to maintain or improve benthic ecosystem health (…)".*

L70 What I'm missing here are explicit reasons 'why' 3D hydrodynamic models may be useful to this topic. The fate of OC was touched on earlier in the introduction but it would be nice to make it clear to the reader here that while some OC will be mineralized by trawling, some will be resuspended and redeposited (I do see it is touched upon later on). Maybe state things about 'how most models do not take into account resuspension and are thus not able to 'track' the fate of OC particles. This study aims to reconcile this through the use of 3D hydrodynamic model.'

*Agreed, this will be added to the introduction.*

Introduction in general: It would be helpful to have some information on potential mechanisms affecting OC dynamics. What causes trawl induced CO2 release? What causes CO2 sequestration in sediments. How does trawling potentially increase mineralization (like though O2 exposure from resuspension) or benthic mineralization? What is the role of macrofauna on these processes?

*Added: "The premise of those studies is that the remineralization of OC to $CO_2$ through respiration by benthic biota is inhibited so long as the OC is trapped in sediment layers under low-oxygen conditions, and that mechanical disturbance will increase oxygenation of that OC, thereby causing a net increase in subaqueous $CO_2$ emissions from the sediment. "*

Fig 1. Instead of italics, are able to use bold or colored text? That would make the words "pop" out more and make things more obvious to readers.

*Font will be changed to bold.*

Table 1. In De Borger et al., 2021 we always found reduced total remineralization (the paper is mentioned under studies that show 'increased remineralization'). There are a few instances of increased oxic mineralization (also with relative changes) but never an increase in total mineralization in that study. Morys et al., 2021 also showed lower rather than higher mineralization (lower benthic respiration as a proxy) as well as Bradshaw et al., (2021) which was not included in the table.

*Table 1 is meant to describe the overall effect on sediment OC without distinguishing between benthic and water column remineralization. We will change the first impact in the table to "Depletion of surficial sedimentary OC" to clarify this and include the reference to Bradshaw et al. (2021) in the table.*

**Methods**

L116 as the other models were named, can you please also give the name of some more details for this ecosystem model.

> *Details of the ECOSMO model will be added.*

L158 Metiers provide more detail than gear types as they account for differences within gear types (mesh size, target species etc.). Perhaps rephrase as readers might see the first sentence (not differentiating between gear types) and become critical without understanding what a metier is.

> *Rephrased: "As the GFW data does not distinguish between specific trawled gear types, a gear type is assigned to each vessel at the vessel's average position according to the dominant métier defined by Eigaard et al. (2016; data in ICES, 2019). A métier groups ..."*

L246 It's not clear to me that this methodology reflects accurate mixing rates from trawling yet. Is this sentence aiming to state that the mixing coefficients are similar to measured trawled areas? The sentence talks about expected bioturbation which makes me think that the mixing coefficients are just similar to that of high bioturbation by fauna. Skeptical readers may want some more evidence stating how this method of calculating trawl induced mixing reflects real conditions. (Perhaps just rephrasing is necessary).

> *We will split the sentence for clarity. The first part is meant to give the reader an idea of the overall impact expected from trawl mixing by comparing it to a natural process.*
>
> *A validation of this approach is complicated both by the scale problem explained in the first paragraph of the subsection, and because, to our knowledge, trawl mixing has never been quantified based on in-situ measurements. Therefore, we compare our mixing coefficients to those found in a sediment core of a heavily trawled area (Spiegel et al., 2023), where the authors argue that bioturbation cannot account for the strong mixing rate and attribute it to trawling. This explanation will be added to the text.*

L305 Experts will understand the logic for looking only at oxic mineralization as this increases relative to anoxic mineralization after trawling. Many readers may not know that so perhaps add a sentence stating that and why anoxic mineralization is not taken into account.

> *Added: "..., assuming that aerobic microbial respiration is the dominant process for OC remineralization when in contact with oxygenated water."*

Methods general: Perhaps I missed it but what I failed to find was how you explicitly account for changes in mineralization in sediments that were trawled. You may have a relative increase in oxic mineralization but we have found that the total mineralization decreases in sediments after trawling as OM is removed from the system. Deposition of OM from trawling may increase total mineralization as it introduces new OM to a system (though the the opposite can also happen if it smothers benthic fauna). Mineralization in the water column also increases. I see that Table 1 shows the effects considered but it's not clear to me how reduced respiration and increased mineralization are specifically incorporated.

*The methods section will be adapted to explain this more clearly. Most of the effects mentioned occur in the model: Macrobenthic respiration scales with biomass, so benthos depletion will reduce that respiration rate. Trawling resuspension temporarily increases water column mineralization due to higher OC content in the water, and can decrease bulk benthic mineralization if the leftover benthic OC is of lower lability. Resuspended labile OC tends to degrade quickly, so even if some of it is redeposited, its effect on benthic rates is not obvious in the model. (See also our response to the previous comment on Table 1). Note that these effects were not explicitly prescribed, but occurred "automatically" as a consequence of the trawling impacts in the process-based model.*

Also, I'm not sure where you link the resuspended sediment from trawling to the hydrodynamic model to see where resuspended sediment and OC ends up. It seems like the model takes into account natural resuspension but the link with trawling resuspension is not clear to me.

*This link is detailed in 2.3.1. Sediment resuspension: "The daily trawling resuspension rate calculated according to Eq. ( 3 ) is added to the natural hydrodynamic resuspension rate at each model time step. (…) The resuspended sediment is distributed evenly over the bottom layer of entire grid cell, where it can be mixed upwards by turbulence and advected horizontally to neighboring grid cells, or redeposited in the absence of currents." We hope that our modifications to the introduction will clarify this approach.*

**Results**

L355 As dead benthos end up in the pool of OC since they're not "removed" from fishing (fishers only remove target fish biomass), how is this accounted for in the results?

*Dead macrobenthos is not added to the OC pool in the model, the implicit assumption being that macrobenthos is quickly degraded by microbial activity. The conceptual difficulty with the treatment of dead macrobenthos in the model is that the other POC source (i.e. comparatively small particulate detritus) is assumed to behave similarly to sediment particles and can be treated as such in the model, whereas macrobenthos carcasses would behave very differently (e.g. they would be consumed by scavengers, would not be resuspended as easily nor mixed downward by bioturbators as effectively). This simplification will be added to the methods and its discussion to the section on model limitations.*

L356: The loss of 14% of benthos in the North Sea can be taken out of context here. Maybe state that this is the difference between trawling and no trawling scenarios.

*Replaced "loss" by "difference".*

Use of 'REF' (reference simulation): I see why you have chosen REF/reference as it represents the status quo of trawling in the North Sea. Nevertheless, I was often confused as a reader since most of the time I see this term (reference) used, it is synonymous to 'control' conditions which are typically undisturbed such as in experimental studies. I would consider using a different term like 'SQ' for 'status quo' simulation or 'baseline' simulation (BASE) to not confuse readers in a similar manner.

*We agree that the use of the term "reference" can be confusing here. We will use "baseline" (BASE) instead of "reference" (REF).*

L368: Trawling pressure is highest in the summer therefore their effect is also high. OC influx is also highest in the spring and summer so does that mitigate some of the trawling effect.

*Fresh OC influx from planktonic detritus during the simulation period, which accounts for less than 0.5% of OC stock in the surface 10 cm sediment, is not considered in the model, as explained in sections 2.1 and 4.5. We do acknowledge that this is a shortcoming of our model. Nevertheless, we argue that this should not have a major impact on our results; Seasonal OC deposition occurs initially as a highly porous and soft fluff layer which is unlikely to provide significant mechanical resistance to bottom trawling gear or buffer resuspension and penetration, so it seems appropriate to ignore the fresh deposit and apply the full trawling resuspension rates and gear penetration to the partially consolidated sediment bed. This argument will be added to section 4.5.*

**Discussion**

L427: I imagine this may be quite difficult to incorporate to the models but how would you answer the question about how trawling leads increases in certain types of macrobenthos like benthic scavengers and more r selected species? In Tiano et al., 2020, we speculate that trawling may have led to more large infauna (sediment mixers/bioirrigators) occuring in the Frisian Front as they tend to survive trawling effects by living deeper in the sediment. Potential discussion here, though our example may be a special case.

*We do not explicitly distinguish between different species traits or functional groups in the model and therefore used an averaged depletion rate of 20%. Nevertheless, the median biomass depth $Z_0$ can be used as a simple indicator for the benthic community structure. A small value of $Z_0$ indicates that macrobenthos are concentrated near the sediment-water interface, while an increase of $Z_0$ suggests that macrobenthos tend to live deeper. We do see a deepening of $Z_0$ after trawling, which is supported by the results in Tiano et al. (2020). A Figure E1b will be added to show this and it will be included in results and discussion. Scavengers will also be mentioned in section 4.5.*

L430: So can you say that the results suggest a greater direct effect on benthos rather than OC?

*It is true that redeposition does not mitigate net benthos depletion in the same way as net OC depletion in the model. However, it is difficult to compare measures of sediment OC and macrobenthos biomass objectively, so we'd prefer to leave this interpretation to the reader.*

L456: As OC influx is lower in the winter, are fisheries closures less effective during these times? I was wondering that since MPA's in the low OC Dogger Bank show little effect, perhaps it's the same during the time of year when OC may be lower.

*It is difficult to answer this based on our results since fishing effort is also lowest in winter, and because we do not consider additional OC sedimentation from planktonic detritus in summer. We prefer not to speculate here but will include this point in section 4.5 of the discussion.*

L521-528: I would specify this explanation of the depositional pattern in the models also somewhere in the methods. It is relevant here in the discussion as it explains certain model limitations but I wanted to ask questions on this topic much earlier in the manuscript so an explanation on how the model handles annual OC deposition early in the methods would be nice.

*This explanation will be moved to the methods section 2.1.*

L587: I'm confused now. I thought all 'reference' simulations were the status quo trawled simulations? Please check this and make consistent throughout the paper.

*Correct, this was an oversight. In an earlier stage of the study, we had labelled the "no-trawling" scenario as the "reference". This also caused the sign of the changes plotted in Fig. 6a and Fig. E1 to be reversed, which will be corrected. We will use "baseline" (BASE) instead of "reference" (REF) and check all instances to make it consistent throughout.*

L608: Perhaps you can also (re)state here how much of the North Sea is closed for the different scenarios.

*We agree with this suggestion but prefer to state the resulting redistribution of trawling effort as a more meaningful measure of the impacts on fishing fleets: "Around 30% of recent trawling effort was located inside of each closure area, with the exception of planned Offshore Windfarms, which overlap with only 5% of trawling effort. (…) closing 23% of the North Sea's area to trawling reduced the net impacts of trawling …"*

L610: I agree with you guys. This is a really good study, great job!

*Thank you!*

---

## Author Comment (AC2)

**Reply to RC2** *(Responses in italics)*

**General comment:**

This paper presents data from a 3D coupled numerical model that is used to quantify the major impacts of bottom trawling on organic carbon and macrobenthos stocks in North Sea sediments. The authors simulate six years of trawling activity and consider four management scenarios in which trawling effort is redistributed from areas inside to areas outside of trawling closure zones. Overall, the paper is well written, but in specific sections, the authors' reasoning is difficult to follow because the manuscript lacks of details about how the data has been treated/modelled (please, see some examples in the specific comments). Some of the implications are well sustained by the presented data, but in some sections, the paper becomes speculative. The authors should try to modulate the conveyed messages and not oversell their ideas if they are not well supported by the presented results. Nevertheless, this is a very nice scientific contribution addressing the spatial impacts and the transport and fate of the resuspended C in the heavily trawled North Sea, which could serve as inspiration for similar future modelling exercises - currently lacking in the scientific literature- and that will help us to properly understand bottom trawling impacts at regional scales and to constrain global estimates.

> *Thank you for the constructive comments that have helped us to clarify and improve our manuscript.*
> *We will extend the methods section to include more details on the model description and adapt the results and discussion sections to communicate the limitations of the study more clearly, and to distinguish the explanations of the model results from the conclusions drawn from those results.*

**Specific comments:**

L 95-100: These coupled models account for Hydrodynamics (SCHISM) sediment dynamics (MORSELFE) and for interactions of OC and macrobenthos in the sediment (TOCMAIM), but what about the C remineralization? How it has been addressed? Perhaps something on this regard should be mentioned here, at the beginning of the Methods section.

> *Carbon remineralization is included in the TOCMAIM model. We will add more information about this to section 2.1.*

L 109-110: It is weird to me to see the three different OC pools based on their bioavailability and degradation rates (fresh, semilabile and refractory) being considered as sediment classes, at the same level than inorganic particles (sand, silt, and clay), and see afterwards in Table A1 that all three OC pools, in term of their sediment dynamics properties, have been associated to the silt class. This is mentioned later in the manuscript (L 183) but to follow how the model has dealt with the OC resuspension, perhaps it should be clarified first here.

> *This information will be added to section 2.1 along with a justification for treating OC as a silt-like fraction. This simplification will be added to the Limitations section (4.5) along with the general discussion of uncertainties of sediment dynamics in the model.*

L 140: Mention here from which period the daily time series of trawling effort from the Global Fishing Watch dataset were extracted. It is introduced latter (L 150), but the reader should know it before, otherwise the paper becomes "mysterious".

> *Added (2015-2020).*

L 149: It is unclear why the authors chose the simulation period of 2000-2005 and the daily fields (of GFW trawling effort?) of 2015-2020 averaged; and it is even less clear how the scaling according to the annual historical landings of demersal fish reported in ICES was conducted afterwards. Perhaps it would have been easier and more realistic to simulate the period of 2015-2020 using the GFW trawling effort, since the simulation period of 2000-2005 based on historical landings could be spatially biased (despite the Couce et al. (2020) findings). In any case, this point should be clarified, regardless the reason behind it. Also, later in the paper, the simulation period of 2000-2005 is used as a reference (REF), but perhaps it would be good to mention this already here, and perhaps restate it or in the introduction of the Management scenarios (section 2.4).

> *We agree that the choice of time periods should be more clearly motivated and will modify sections 2.1 and 2.2 to clarify our approach and describe the scaling method more precisely.*
> *"Reference" was not used to indicate the time period, but rather to indicate the simulations using the actual trawling effort (in contrast to the modified effort used for the management scenarios). This will be changed to avoid confusion (see our response to the comment on L 256 below).*
>
> *While we agree that simulating more recent years would be appealing, there are several conceptual and technical reasons why we chose 2000-2005 as the study period, which we will make clearer in the manuscript:*
>
> 1. *Fishing effort estimated by GFW increases from year to year, but we believe that this increase does not reflect an increase in actual fishing effort. Instead, this effect is caused by the increasing coverage of AIS-based vessel data. This is evidenced by the comparison to the (VMS-based) ICES data, showing a large underestimation in earlier years and subsequent gradual convergence with the ICES effort. Therefore, using the GFW data "as-is" would introduce a strong bias. Our method of averaging and scaling is meant to more closely match the ICES effort.*
> 2. *Whiereas fishing pressure has remained relatively stable in recent years, the chosen model period 2000-2005 contains strong interannual variations in fishing effort in the North Sea, which allows to more clearly discern the effects of different levels of fishing effort.*
> 3. *The atmospheric forcing dataset (Geyer, 2017), with which the model has been validated (Kossack et al., 2023), does not currently extend past 2018. There are other atmospheric models that could be used, but this would require complete re-calibration the model, as hydrodynamic models are sensitive to changes in atmospheric forcing.*

L 183-185: In the paper, only the silt content is considered to estimate the resuspension rate of trawlers, but, generally, OC content increases in muddy sediments that are finer than silts. What about the clay fraction? Something should be mentioned on this regard.

*The term "silt fraction" is perhaps misleading here, since in the original formula of O'Neill and Summerbell (2016) for trawling resuspension, "silt fraction" encompassed all sediments <63 μm in diameter, including the clay fraction, i.e. the total "mud content". The mud content is also what we use in our estimate of trawling resuspension, so it is consistent with the resuspension formulation. The phrasing will be modified to clarify this. Regarding OC content, note that it is set independently of grain size according to sediment maps (Bockelmann et al., 2018), though it is true that OC content tends to increase with finer and more cohesive particles.*

L 256: Define here what is exactly the reference simulation (REF). This is the first time that this acronym is used.

*Rephrased for clarification: "Six simulations are carried out for 2000-2005 using different distributions of trawling pressure: a baseline simulation (BASE) using the actual trawling distribution and representing the status quo, serving as a reference to which the remaining scenarios can be compared, one scenario without ..."*

*Note that in accordance with the other reviewer's comments regarding the potentially misleading use of the term "reference", we will change it to "baseline" (BASE).*

L 340-344: The Figure 5, illustrating the change in average trawling resuspension and erosion rates, is hard to follow if previously the corresponding maps of redistribution of trawling for each scenario (similar to the map in Figure 3) are not shown. Otherwise, the authors skip the illustration of one critical step of the computing process, which is the redistribution of trawling effort. The limits outlining the areas of trawling closure zones are hard to discern, and in some maps it is difficult to identify if the areas are inside or outside the lines. Perhaps the areas should be filled with a hatched pattern. Additionally, the time constrain is not mentioned in the figure caption.

*We will replace Fig. 5 with a figure showing the trawling effort for the scenarios and indicate the closure areas more clearly.*

L 390-394: The same as in Figure 5. It t would be desirable to see the trawling effort maps before presenting the average differences of changes in OC fluxes in Figure 7, and also the areas of trawling closure zones should be hatched, since they are hard to discern.

*We will indicate the closure areas in Fig. 5 and Fig. 7 more clearly.*

L 430-432: This sentence needs a proper reference to support such a strong statement. This is an example of the speculative sentences found throughout the text.

*This sentence is meant to explain what is occurring in the model, but was not intended as a general claim. Several sentences in the discussion and conclusion sections will be re-phrased to distinguish between explanation of the model results from the conclusions drawn from those results more clearly.*

L 444-445: Macrobenthos biomass responses are shown for the first time in the Discussion section, while they should be included in the Results section.

*The results for biomass will be included in the results section.*

L 516-579: I miss the Model limitations subsection some paragraph dealing with the need to improve the computation of the sediment and C resuspension, transport and re-deposition processes, which have been treated quite simplistically in this modelling exercise. To me, this is a key aspect, since most of the distribution maps and the computation of the C fluxes caused by trawling largely depend on this parametrization. Besides, several of the listed "limitations" on this subsection are not inherent of the models used (SCHISM, MORSELFE and TOCMAIM), but instead they are aspects that could not considered or addressed in the paper using these models. Perhaps the title of this subsection could be renamed as "Model limitations and unaddressed processes/mechanisms".

*We will add a paragraph discussing the simplification of sediment dynamics in the model and associated and uncertainty and renamed the subsection to "Study limitations".*

L 582: Define the period during which the daily time series were generated.

*Added (2015-2020)*

L 585: Again, the period of the six consecutive years is missing.

*Added (2000-2005)*

L 706: There is no mention to the availability of the code of the MORSELFE model.

*MORSELFE will be added to the code availability section. It is integrated as the sediment module within SCHISM, so it is available at the same source.*

---

## Author Response (AR1)

**Reply to RC1** *(Responses in italics, changes to manuscript in red)*

**General comments**

Porz et al., conducted a modelling study which uses coupled 3D hydrodynamic simulations with a bioturbation model to look at trawling impacts in the North Sea. I think this may be the most holistic and up to date research on this topic that I've seen. Most studies on this topic have just looked at the direct impact of trawling on sediment carbon within a trawl track and some look at how that effect might evolve through time. This study looks at the potential fate of OC once resuspended and shows how trawling can lead to lower OC in some areas and more OC in other areas (something observed in field observations before as well). Furthermore, I was happy to see that in the simulations regarding fisheries closures, trawling effort was redistributed rather than assumed to have decreased. All the praise aside, I think the manuscript still has areas of improvement, most of which I've outlined below. My main concern is that, although, I've seen it mentioned throughout the paper, it was never clear to me exactly how remineralization was addressed by the model. For me, the parts about the removal of OC through resuspension and redistribution to other areas was better explained but I would like to know exactly in the methods, your assumptions on benthic mineralization after trawling as well as once resuspended. It is stated in the discussion that OC is degraded once resuspended but I would like to see further details behind this like how much mineralization decreased/increased (both in the water column and in the sediment) and why.

*Thank you for the constructive comments that have helped us to clarify and improve our manuscript.*

*We have extended the methods section to include more details on the model description and explained the mechanisms behind trawling resuspension on OC in the model more clearly in the results.*

*Added to 2.1: "Within the sediment, OC is remineralized by macrobenthic uptake, which also scales with biomass, and by microbial degradation. Microbial OC remineralization rates decrease with sediment depth to account for reduction in microbial activity with decreasing oxygen availability, leading to slower degradation in deeper sediment layers. The first-order (oxic) remineralization rates are applied to OC suspended in the water column and within the uppermost sediment layer. For further details on the TOCMAIM model, the reader is referred to Zhang and Wirtz (2017)."*

*Added to 4.1: "Trawling resuspension is a primary mechanism leading to higher OC remineralization in the model, as resuspended sediment is remineralized most rapidly in the water column. In addition, the sediment-water interface is shifted downward wherever it is eroded by trawling, exposing more benthic OC to the oxic remineralization rate."*

**Detailed comments**

L11 There are reasons why 3D hydrodynamic models provide much needed insight on this topic but this is not yet clear to the reader. It would be nice to transition from the debate amongst scientists to something like "However, current discussions around the fate of resuspended organic matter are lacking. To help resolve this, we used 3D hydrodynamic… "

*Added to the abstract: "An issue that has remained unaddressed thus far regards the fate of organic carbon resuspended into the water column following disturbance by fishing gear. To resolve this,…"*

L25 is there anything that could be said about what makes these sediments vulnerable? High labile OC/biomass, low natural disturbance etc.?

*Added to the abstract: „The largest positive impact arose for trawling closures in Carbon Protection Zones, which were defined as areas where organic carbon is both plentiful and labile, and thereby most vulnerable to disturbance."*

L30 so does this give reason to advocate for spatial as opposed to effort management?

*In the context of carbon and habitat protection, our results may be used to argue in this direction. However, this is only one consideration for fisheries management, which has mostly prioritized prevention of overfishing. In addition, our results do not account for possible changes in catches following spatial effort redistribution. To prevent misunderstandings and considering the other reviewer's caution not to overstate our results, we prefer not to advocate for specific management strategies here.*

L49 "seabed destruction" is vague and perhaps too loaded of a term here. Can you be more specific?

*Rephrased to "Efforts to maintain or improve benthic ecosystem health (…)".*

L70 What I'm missing here are explicit reasons 'why' 3D hydrodynamic models may be useful to this topic. The fate of OC was touched on earlier in the introduction but it would be nice to make it clear to the reader here that while some OC will be mineralized by trawling, some will be resuspended and redeposited (I do see it is touched upon later on). Maybe state things about 'how most models do not take into account resuspension and are thus not able to 'track' the fate of OC particles. This study aims to reconcile this through the use of 3D hydrodynamic model.'

*Added to the introduction: "The use of a 3D model enables tracking the fate of particles in space and time through multiple cycles of resuspension and transport in the water column until eventual redeposition and ultimate burial."*

Introduction in general: It would be helpful to have some information on potential mechanisms affecting OC dynamics. What causes trawl induced CO2 release? What causes CO2 sequestration in sediments. How does trawling potentially increase mineralization (like though O2 exposure from resuspension) or benthic mineralization? What is the role of macrofauna on these processes?

*Added to the introduction: "The premise of those studies is that the remineralization of OC to $CO_2$ through respiration by benthic biota is inhibited so long as the OC is trapped in sediment layers under low-oxygen conditions, and that mechanical disturbance will increase oxygenation of that OC, thereby causing a net increase in subaqueous $CO_2$ emissions from the sediment."*

Fig 1. Instead of italics, are able to use bold or colored text? That would make the words "pop" out more and make things more obvious to readers.

*Font in Fig. 1 changed to bold.*

Table 1. In De Borger et al., 2021 we always found reduced total remineralization (the paper is mentioned under studies that show 'increased remineralization'). There are a few instances of increased oxic mineralization (also with relative changes) but never an increase in total mineralization in that study. Morys et al., 2021 also showed lower rather than higher mineralization (lower benthic respiration as a proxy) as well as Bradshaw et al., (2021) which was not included in the table.

*Table 1 is meant to describe the overall effect on sediment OC without distinguishing between benthic and water column remineralization.*

*First impact in Table 1 changed to "Depletion of surficial sedimentary OC" and included the reference to Bradshaw et al. (2021) in the table.*

**Methods**

L116 as the other models were named, can you please also give the name of some more details for this ecosystem model.

*Added to 2.1: "For this uncoupled simulation, input of OC at the sediment surface in the form of planktonic detritus is assigned according to the outputs of an NPZD-type ecosystem model (ECOSMO; Daewel and Schrum, 2013), which calculates deposition patterns as governed by ecosystem production and hydrodynamics following phytoplankton blooms."*

L158 Metiers provide more detail than gear types as they account for differences within gear types (mesh size, target species etc.). Perhaps rephrase as readers might see the first sentence (not differentiating between gear types) and become critical without understanding what a metier is.

*Rephrased: "As the GFW data does not distinguish between specific trawled gear types, a gear type is assigned to each vessel at the vessel's average position according to the dominant métier defined by Eigaard et al. (2016; data in ICES, 2019). A métier groups ..."*

L246 It's not clear to me that this methodology reflects accurate mixing rates from trawling yet. Is this sentence aiming to state that the mixing coefficients are similar to measured trawled areas? The sentence talks about expected bioturbation which makes me think that the mixing coefficients are just similar to that of high bioturbation by fauna. Skeptical readers may want some more evidence stating how this method of calculating trawl induced mixing reflects real conditions. (Perhaps just rephrasing is necessary).

*The first part is meant to give the reader an idea of the overall impact expected from trawl mixing by comparing it to a natural process.*

*We have split the sentence for clarity: "This is in the same order of magnitude as expected natural bioturbation intensities in the North Sea (Teal et al., 2008). However,*

*in heavily trawled areas such as the Skagerrak, estimated trawl mixing reaches magnitudes on the order of 0.1–1 cm2 d-1 , exceeding expected bioturbation (see Figure C2).*

*To our knowledge, trawl mixing has never been quantified based on in-situ measurements, complicating a validation of our approach. Nevertheless, our estimates are supported by Spiegel et al. (2023), who attributed an exceptionally strong and deep mixing signal in a sediment sample retrieved from the Skagerrak to mixing by bottom trawling. They estimated mixing rates of more than 0.1 cm2 d-1 at chronically trawled sites, more than twice as high as at comparable untrawled sites and similar to our mixing estimates in that region.”*

L305 Experts will understand the logic for looking only at oxic mineralization as this increases relative to anoxic mineralization after trawling. Many readers may not know that so perhaps add a sentence stating that and why anoxic mineralization is not taken into account.

*Added: “…, assuming that aerobic microbial respiration is the dominant process for OC remineralization when in contact with oxygenated water.”*

Methods general: Perhaps I missed it but what I failed to find was how you explicitly account for changes in mineralization in sediments that were trawled. You may have a relative increase in oxic mineralization but we have found that the total mineralization decreases in sediments after trawling as OM is removed from the system. Deposition of OM from trawling may increase total mineralization as it introduces new OM to a system (though the the opposite can also happen if it smothers benthic fauna). Mineralization in the water column also increases. I see that Table 1 shows the effects considered but it's not clear to me how reduced respiration and increased mineralization are specifically incorporated.

*Most of the effects mentioned occur in the model: Macrobenthic respiration scales with biomass, so benthos depletion will reduce that respiration rate. Trawling resuspension temporarily increases water column mineralization due to higher OC content in the water, and can decrease bulk benthic mineralization if the leftover benthic OC is of lower lability. Resuspended labile OC tends to degrade quickly, so even if some of it is redeposited, its effect on benthic rates is not obvious in the model. (See also our response to the previous comment on Table 1). Note that these effects were not explicitly prescribed, but occurred “automatically” as a consequence of the trawling impacts in the process-based model.*

*The methods section has been adapted to explain this more clearly (see above for changes).*

Also, I'm not sure where you link the resuspended sediment from trawling to the hydrodynamic model to see where resuspended sediment and OC ends up. It seems like the model takes into account natural resuspension but the link with trawling resuspension is not clear to me.

*This link is detailed in 2.3.1. Sediment resuspension: “The daily trawling resuspension rate calculated according to Eq. ( 3 ) is added to the natural hydrodynamic resuspension rate at each model time step. (…) The resuspended sediment is distributed evenly over the bottom layer of entire grid cell, where it can be mixed upwards by turbulence and advected horizontally to neighboring grid cells, or*

*redeposited in the absence of currents." We hope that our modifications to the introduction have clarified this approach.*

**Results**

L355 As dead benthos end up in the pool of OC since they're not "removed" from fishing (fishers only remove target fish biomass), how is this accounted for in the results?

> *Dead macrobenthos is not added to the OC pool in the model, the implicit assumption being that macrobenthos is quickly degraded by microbial activity.*
>
> *Added to 2.3.3: "Though dead macrobenthos can be considered part of the sediment OC pool, this term is ignored in the model, assuming that dead macrobenthos is quickly degraded by microbial activity. If composed entirely of labile OC, dead macrobenthos would be degraded down to less than 10% of its initial mass within 42 days after depletion by a trawl in the model."*
>
> *Added to 4.5: "Another simplification concerns the treatment of dead macrobenthos following depletion by trawling. The conceptual difficulty is that the other OC source (i.e. comparatively small particulate detritus) is assumed to behave similarly to sediment particles and can be treated as such in the model, whereas macrobenthos carcasses would behave very differently, e.g. they would be consumed by scavengers, would not be resuspended as easily or mixed downward by bioturbators as effectively. Though treating depleted macrobenthos an additional source of OC may initially offset our estimated net impacts of trawling to some degree, we consider that this effect should decrease with time as the benthic community structures adjust to the disturbed habitat."*

L356: The loss of 14% of benthos in the North Sea can be taken out of context here. Maybe state that this is the difference between trawling and no trawling scenarios.

> *Replaced "loss" by "difference".*

Use of 'REF' (reference simulation): I see why you have chosen REF/reference as it represents the status quo of trawling in the North Sea. Nevertheless, I was often confused as a reader since most of the time I see this term (reference) used, it is synonymous to 'control' conditions which are typically undisturbed such as in experimental studies. I would consider using a different term like 'SQ' for 'status quo' simulation or 'baseline' simulation (BASE) to not confuse readers in a similar manner.

> *We agree that the use of the term "reference" can be confusing here.*
>
> *Changed all instances of "reference" (REF) to "baseline" (BASE).*

L368: Trawling pressure is highest in the summer therefore their effect is also high. OC influx is also highest in the spring and summer so does that mitigate some of the trawling effect.

> *Fresh OC influx from planktonic detritus during the simulation period, which accounts for less than 0.5% of OC stock in the surface 10 cm sediment, is not considered in the model, as explained in sections 2.1 and 4.5. We do acknowledge that this is a*

*shortcoming of our model. Nevertheless, we argue that this should not have a major impact on our results.*

*Added to 4.5: "Moreover, seasonal OC deposition occurs initially as a low-density fluff layer (Jago and Jones, 1998; Beaulieu, 2002), which is unlikely to provide significant mechanical resistance to bottom trawling gear, so it seems appropriate to apply the full trawling resuspension rates and gear penetration to the existing, consolidated sediment bed. Any further addition of fresh OC to the sediment surface is therefore not expected to change the findings significantly, though this shall be confirmed by longer-term simulations in which ecosystem production is included."*

**Discussion**

L427: I imagine this may be quite difficult to incorporate to the models but how would you answer the question about how trawling leads increases in certain types of macrobenthos like benthic scavengers and more r selected species? In Tiano et al., 2020, we speculate that trawling may have led to more large infauna (sediment mixers/bioirrigators) occuring in the Frisian Front as they tend to survive trawling effects by living deeper in the sediment. Potential discussion here, though our example may be a special case.

*We do not explicitly distinguish between different species traits or functional groups in the model and therefore used an averaged depletion rate of 20%. Nevertheless, the median biomass depth $Z_0$ can be used as a simple indicator for the benthic community structure. A small value of $Z_0$ indicates that macrobenthos are concentrated near the sediment-water interface, while an increase of $Z_0$ suggests that macrobenthos tend to live deeper. We do see a deepening of $Z_0$ after trawling, which is supported by the results in Tiano et al. (2020).*

*Figure E1b has been added to show this and it is now included in results and discussion. Scavengers are now also mentioned in 4.5.*

L430: So can you say that the results suggest a greater direct effect on benthos rather than OC?

*It is true that redeposition does not mitigate net benthos depletion in the same way as net OC depletion in the model. However, it is difficult to compare measures of sediment OC and macrobenthos biomass objectively, so we'd prefer to leave this interpretation to the reader.*

L456: As OC influx is lower in the winter, are fisheries closures less effective during these times? I was wondering that since MPA's in the low OC Dogger Bank show little effect, perhaps it's the same during the time of year when OC may be lower.

*It is difficult to answer this based on our results since fishing effort is also lowest in winter, and because we do not consider additional OC sedimentation from planktonic detritus in summer.*

*Added to 4.5: "Such coupled simulations may also be used to investigate whether trawling closures are more effective in some seasons than in others."*

L521-528: I would specify this explanation of the depositional pattern in the models also somewhere in the methods. It is relevant here in the discussion as it explains certain model limitations but I wanted to ask questions on this topic much earlier in the manuscript so an explanation on how the model handles annual OC deposition early in the methods would be nice.

*Explanation moved to 2.1: "… deposition patterns as governed by ecosystem production and hydrodynamics following phytoplankton blooms."*

L587: I'm confused now. I thought all 'reference' simulations were the status quo trawled simulations? Please check this and make consistent throughout the paper.

*Correct, this was an oversight. In an earlier stage of the study, we had labelled the "no-trawling" scenario as the "reference". This also caused the sign of the changes plotted in Fig. 6a and Fig. E1 to be reversed. We have used "baseline" (BASE) instead of "reference" (REF) and checked all instances to make it consistent throughout.*

*Sign of changes in Fig. 6a and Fig. E1 corrected.*

*All instances of "reference" (REF) changed to "baseline" (BASE).*

L608: Perhaps you can also (re)state here how much of the North Sea is closed for the different scenarios.

*We agree with this suggestion but prefer to state the resulting redistribution of trawling effort as a more meaningful measure of the impacts on fishing fleets.*

*Added to section 5: "About 28–29% of recent trawling effort was located inside each of the closure areas, with the exception of planned Offshore Windfarms, which overlap with only 5% of recent trawling effort. […] closing 23% of the North Sea's area to trawling reduced the net impacts of trawling on organic carbon by 29% and on macrobenthos biomass by 54% […]"*

L610: I agree with you guys. This is a really good study, great job!

*Thank you!*

**Reply to RC2** *(Responses in italics, changes to manuscript in red)*

**General comment:**

This paper presents data from a 3D coupled numerical model that is used to quantify the major impacts of bottom trawling on organic carbon and macrobenthos stocks in North Sea sediments. The authors simulate six years of trawling activity and consider four management scenarios in which trawling effort is redistributed from areas inside to areas outside of trawling closure zones. Overall, the paper is well written, but in specific sections, the authors' reasoning is difficult to follow because the manuscript lacks of details about how the data has been treated/modelled (please, see some examples in the specific comments). Some of the implications are well sustained by the presented data, but in some sections, the paper becomes speculative. The authors should try to modulate the conveyed messages and not oversell their ideas if they are not well supported by the presented results. Nevertheless, this is a very nice scientific contribution addressing the spatial impacts and the transport and fate of the resuspended C in the heavily trawled North Sea, which could serve as inspiration for similar future modelling exercises -currently lacking in the scientific literature- and that will help us to properly understand bottom trawling impacts at regional scales and to constrain global estimates.

> *Thank you for the constructive comments that have helped us to clarify and improve our manuscript.*
> *We extended the methods section to include more details on the model description and adapt the results and discussion sections to communicate the limitations of the study more clearly, and to distinguish the explanations of the model results from the conclusions drawn from those results.*

**Specific comments:**

L 95-100: These coupled models account for Hydrodynamics (SCHISM) sediment dynamics (MORSELFE) and for interactions of OC and macrobenthos in the sediment (TOCMAIM), but what about the C remineralization? How it has been addressed? Perhaps something on this regard should be mentioned here, at the beginning of the Methods section.

> *Carbon remineralization is included in the TOCMAIM model. We have added more information about this to section 2.1. (see response to RC1 for changes)*

L 109-110: It is weird to me to see the three different OC pools based on their bioavailability and degradation rates (fresh, semilabile and refractory) being considered as sediment classes, at the same level than inorganic particles (sand, silt, and clay), and see afterwards in Table A1 that all three OC pools, in term of their sediment dynamics properties, have been associated to the silt class. This is mentioned later in the manuscript (L 183) but to follow how the model has dealt with the OC resuspension, perhaps it should be clarified first here.

> *Added to 2.1: "Organic carbon is usually adsorbed to fine-grained sediment (silt and clay), and presence of OC typically causes the formation of relatively stable, low-density microflocs (e.g. Virto et al., 2008) which we assume to behave similarly to silt-sized particles. Therefore, the three sediment classes representing OC are treated identically to the inorganic silt class regarding their sediment dynamic properties."*

*This simplification was added to the Limitations along with the general discussion of uncertainties of sediment dynamics in the model.*

*Added to 4.5: "A significant limitation of this study concerns the simplified treatment of sediment dynamics in the model. The parametrizations of resuspension, settling and deposition do not account for more complex processes affecting cohesive sediment such as flocculation, hindered settling and consolidation, nor bedload transport. Though parametrizations for such processes have been developed (e.g. Winterwerp, 2002; Sherwood et al., 2018), a lack of observational data for near-bottom suspended sediment concentrations in the North Sea currently prohibits the validation of such dynamics. In general, sediment dynamic models do not benefit from introducing additional complexity without suitable validation data (Arlinghaus et al., 2022). Further observational data and model sensitivity experiments are therefore required to gauge and constrain the validity of our results."*

L 140: Mention here from which period the daily time series of trawling effort from the Global Fishing Watch dataset were extracted. It is introduced latter (L 150), but the reader should know it before, otherwise the paper becomes "mysterious".

*Added (2015-2020).*

L 149: It is unclear why the authors chose the simulation period of 2000-2005 and the daily fields (of GFW trawling effort?) of 2015-2020 averaged; and it is even less clear how the scaling according to the annual historical landings of demersal fish reported in ICES was conducted afterwards. Perhaps it would have been easier and more realistic to simulate the period of 2015-2020 using the GFW trawling effort, since the simulation period of 2000-2005 based on historical landings could be spatially biased (despite the Couce et al. (2020) findings). In any case, this point should be clarified, regardless the reason behind it. Also, later in the paper, the simulation period of 2000-2005 is used as a reference (REF), but perhaps it would be good to mention this already here, and perhaps restate it or in the introduction of the Management scenarios (section 2.4).

*We agree that the choice of time periods should be more clearly motivated and have modified sections 2.1 and 2.2 to clarify our approach and describe the scaling method more precisely.*

*Added to 2.1: "This period was chosen because it contains varying trends in trawling effort, with a moderate increase in demersal fish landings during the first years, followed by a sharp decrease in the later half, after which the levels have remained similar until recent years (ICES, 2023b)."*

*Added to 2.2: "For the simulation period of 2000–2005, the daily fields of 2015–2020 are averaged and hindcast using annual historical landing data of demersal fish reported in ICES (2017). The GFW daily trawled hours averaged over 2015–2020 are about 10% higher than those of 2017, somewhat mitigating the 10% lower effort in the GFW data compared to those in ICES (2019). The daily hindcast in each year $y$ is therefore performed by scaling the averaged daily fields $\bar{f}_{daily}$ of 2015–2020 with the landings in year $y$ with respect to the landings in 2017:*

$$f_{\mathrm{daily},y} = \bar{f}_{\mathrm{daily}} \cdot \frac{\mathrm{landings}_y}{\mathrm{landings}_{2017}}. \qquad (\,1\,)$$

*"Reference" was not used to indicate the time period, but rather to indicate the simulations using the actual trawling effort (in contrast to the modified effort used for the management scenarios). This was changed to avoid confusion (see our response to the comment on L 256 below).*

L 183-185: In the paper, only the silt content is considered to estimate the resuspension rate of trawlers, but, generally, OC content increases in muddy sediments that are finer than silts. What about the clay fraction? Something should be mentioned on this regard.

*The term "silt fraction" is perhaps misleading here, since in the original formula of O'Neill and Summerbell (2016) for trawling resuspension, "silt fraction" encompassed all sediments <63 µm in diameter, including the clay fraction, i.e. the total "mud content". The mud content is also what we use in our estimate of trawling resuspension, so it is consistent with the resuspension formulation.*

*Phrasing modified: "[…] where s_f is the silt mud content of the seabed (proportion of silt- and clay-sized particles) and […]"*

*Regarding OC content, note that it is set independently of grain size according to sediment maps (Bockelmann et al., 2018), though it is true that OC content tends to increase with finer and more cohesive particles.*

L 256: Define here what is exactly the reference simulation (REF). This is the first time that this acronym is used.

*Rephrased for clarification: "Six simulations are carried out for 2000-2005 using different distributions of trawling pressure: a baseline simulation (BASE) using the actual trawling distribution and representing the status quo, serving as a reference to which the remaining scenarios can be compared, one scenario without …"*

*Note that in accordance with the other reviewer's comments regarding the potentially misleading use of the term "reference", we changed it to "baseline" (BASE).*

L 340-344: The Figure 5, illustrating the change in average trawling resuspension and erosion rates, is hard to follow if previously the corresponding maps of redistribution of trawling for each scenario (similar to the map in Figure 3) are not shown. Otherwise, the authors skip the illustration of one critical step of the computing process, which is the redistribution of trawling effort. The limits outlining the areas of trawling closure zones are hard to discern, and in some maps it is difficult to identify if the areas are inside or outside the lines. Perhaps the areas should be filled with a hatched pattern. Additionally, the time constrain is not mentioned in the figure caption.

*Fig. 5 replaced with a plot showing the trawling effort for the scenarios and indicate the closure areas more clearly.*

L 390-394: The same as in Figure 5. It t would be desirable to see the trawling effort maps before presenting the average differences of changes in OC fluxes in Figure 7, and also the areas of trawling closure zones should be hatched, since they are hard to discern.

*Closure areas in Fig. 5 and Fig. 7 indicated more clearly.*

L 430-432: This sentence needs a proper reference to support such a strong statement. This is an example of the speculative sentences found throughout the text.

*This sentence is meant to explain what is occurring in the model, but was not intended as a general claim.*

*Re-phrased several sentences in the discussion and conclusion sections to distinguish between explanation of the model results and the conclusions drawn from those results more clearly.*

L 444-445: Macrobenthos biomass responses are shown for the first time in the Discussion section, while they should be included in the Results section.

*Moved results for biomass to the results section.*

L 516-579: I miss the Model limitations subsection some paragraph dealing with the need to improve the computation of the sediment and C resuspension, transport and re-deposition processes, which have been treated quite simplistically in this modelling exercise. To me, this is a key aspect, since most of the distribution maps and the computation of the C fluxes caused by trawling largely depend on this parametrization. Besides, several of the listed "limitations" on this subsection are not inherent of the models used (SCHISM, MORSELFE and TOCMAIM), but instead they are aspects that could not considered or addressed in the paper using these models. Perhaps the title of this subsection could be renamed as "Model limitations and unaddressed processes/mechanisms".

*Added a paragraph discussing the simplification of sediment dynamics in the model and associated and uncertainty (see above).*

*Renamed the subsection to "Study limitations".*

L 582: Define the period during which the daily time series were generated.

*Added (2015-2020)*

L 585: Again, the period of the six consecutive years is missing.

*Added (2000-2005)*

L 706: There is no mention to the availability of the code of the MORSELFE model.

*MORSELFE added to the code availability section.*

*It is integrated as the sediment module within SCHISM, so it is available at the same source.*